# Hydralazine targets cAMP-dependent protein kinase leading to sirtuin1/5 activation and lifespan extension in *C. elegans*

Esmaeil Dehghan [1]*, Mohammad Goodarzi[1], Bahar Saremi[2], Rueyling Lin[3] & Hamid Mirzaei [1]*

Therapeutic activation of mitochondrial function has been suggested as an effective strategy to combat aging. Hydralazine is an FDA-approved drug used in the treatment of hypertension, heart failure and cancer. Hydralazine has been recently shown to promote lifespan in *C. elegans*, rotifer and yeast through a mechanism which has remained elusive. Here we report cAMP-dependent protein kinase (PKA) as the direct target of hydralazine. Using in vitro and in vivo models, we demonstrate a mechanism in which binding and stabilization of a catalytic subunit of PKA by hydralazine lead to improved mitochondrial function and metabolic homeostasis via the SIRT1/SIRT5 axis, which underlies hydralazine's prolongevity and stress resistance benefits. Hydralazine also protects mitochondrial metabolism and function resulting in restoration of health and lifespan in *C. elegans* under high glucose and other stress conditions. Our data also provide new insights into the mechanism(s) that explain various other known beneficial effects of hydralazine.

[1] Department of Biochemistry, UT Southwestern Medical Center, Dallas, TX 75390, USA. [2] University of Texas at Arlington and University of Texas Southwestern Medical Center Joint Biomedical Engineering Program, Arlington, TX 75390, USA. [3] Department of Molecular Biology, UT Southwestern Medical Center, Dallas, TX, USA. *email: Esmaeil.Dehghan@UTSouthwestern.edu; Hamid.Mirzaei@UTsouthwestern.edu

The number of senior citizens over the age of 80 will triple in the United States by the year 2050. If debilitating aging pathologies are not appropriately managed this shift in population average age will come at a high cost, socially and economically[1]. Aging involves a series of interconnected pathologies and studies indicate that delaying one age-related pathology may stave off others. In other words, extending lifespan seems to require an extension of healthspan[1]. Therefore, developing pharmaceutical interventions that decelerate aging pathologies are highly desirable[2].

As a dynamic network of double membrane-bound organelles, mitochondria house central energy metabolism including machinery powering the tricarboxylic acid (TCA) cycle and oxidative phosphorylation (OxPhos)[3]. Additionally, mitochondria are critical mediators of cell signaling and apoptosis and are involved in calcium buffering[4,5]. As cells age, the efficiency of the respiratory chain diminishes resulting in a reduction of ATP generation, increased electron leakage from the mitochondrial electron transfer chain (ETC), and elevated oxidative stress[5]. Hence, restoration of mitochondrial performance is critical for the deceleration of aging and age-related disorders.

The mammalian sirtuin family of proteins (SIRT1-SIRT7) has been the focus of many studies for their perceived regulatory role in a wide variety of cellular processes, including cellular metabolism and aging[6]. A large body of evidence links sirtuins, particularly SIRT1, to aging and age-related diseases[7,8]. Transgenic mice overexpressing Sirt1 in brain live longer, maintain youthful mitochondrial morphology/function in skeletal muscles and exhibit decelerated aging[9]. SIRT1 and mitochondrial sirtuins (SIRT3-SIRT5) appear to play pivotal roles in maintaining mitochondrial function, and their age-related decline correlates with the pathophysiology of aging[10].

Cyclic AMP (cAMP), one of the most versatile second messenger molecules, plays critical roles in many biological processes. cAMP has been reported to increase sirtuin levels and delay the onset of age-related pathologies by mimicking the effects of calorie restriction (CR) in mice[11]. cAMP-dependent kinase (PKA), consisting of two regulatory and two catalytic subunits, is the primary downstream target of the cAMP signaling pathway. Binding of cAMP to PKA regulatory subunits induces a conformational change that results in the release and activation of the catalytic subunits and phosphorylation of hundreds of substrates involved in the regulation of myriad cellular signaling pathways[12]. cAMP-induced PKA activation results in phosphorylation and activation of SIRT1 which in turn modulates mitochondrial function and fatty acid oxidation[13]. It has also been shown that the PKA catalytic subunit decreases with aging while acute activation of the cAMP/PKA pathway in aging Drosophila promotes axonal transport of mitochondria in neurons[14]. Consequently, modulation of cAMP/PKA signaling seems to be a promising strategy for the activation of sirtuins and improvement of mitochondrial function in aging organisms.

Repurposing of existing FDA approved drugs is a cost-effective strategy for new therapy development[15]. The FDA approved hydralazine in 1953, and because of its effectiveness and safety, it is still prescribed[16]. In addition to its application in the treatment of carbonyl-mediated pathologies, hydralazine was repurposed in the 1980s for the treatment of heart failure and again in the 2000s for cancer epigenetic therapy[17]. More recently we demonstrated that hydralazine activates the NRF2/SKN-1 pathway and extends C. elegans lifespan[18]. The anti-aging benefits of hydralazine have also been demonstrated in rotifers in a screen for identification of life-extending FDA approved drugs[15,16]. Hydralazine ameliorates behavioral disorders and prevents loss of dopaminergic neurons in the substantia nigra (SN) and striatum by the activation of Nrf2-ARE pathway in an MPTP (1-methyl-4-phenyl-1,2,3,6-tetrahydropyridine)-induced mouse model of Parkinson's disease[19]. Despite numerous studies, the fundamental mechanism of hydralazine's action is poorly understood. In this report, we identified PKA as direct target of hydralazine, which activates a SIRT1/SIRT5 axis to promote mitochondrial function and confer health and pro-longevity benefits.

## Results

**Hydralazine improves mitochondrial function**. To study the effect of hydralazine on the mitochondrial function, we measured different markers of mitochondrial activity and biogenesis using two different cell lines; human neuroblastoma SH-SY5Y and mouse myoblast C2C12 cells. SH-SY5Y cells were treated for 72 h in DMEM containing different concentrations of hydralazine, resveratrol as a positive control, and isoniazid as a negative control. The mitochondrial membrane potential ($\Delta\psi$m) was assessed by staining the cells with tetramethylrhodamine ethyl ester (TMRE) to stain active mitochondria (Fig. 1a). These data demonstrate a dose-dependent increase in the mitochondrial membrane potential with hydralazine treatment. We next investigated mitochondrial biogenesis by measuring the ratio between mitochondrial DNA and nuclear DNA (mtDNA/nDNA), and by measuring mitochondrial mass. Quantitative PCR measurement of NADH dehydrogenase subunit 5 (MT-ND5) and mitochondrial encoded ribosomal RNAs (MT-RNR) in DNA extracted from hydralazine-treated cells (5 μM for 72 h) showed an increase in the mtDNA/nDNA ratio (Fig. 1b). Mitochondrial mass was evaluated by Western blot analysis of cytochrome c (CYCS). In SH-SY5Y cells, a dose-dependent increase in CYCS protein abundance was observed with hydralazine treatment (Fig. 1c).

To evaluate mitochondrial function in cells treated with hydralazine, we examined the activity of the individual complexes of the mitochondrial ETC (complex I, II, and IV) as well as ATP synthase complex V in C2C12 cells treated with different concentrations of hydralazine for three days. Spectrophotometric analysis showed a general activation of mitochondrial ETC complexes (Fig. 1d–g). Furthermore, consistent with the spectrophotometric data, in-gel activity assays of mitochondrial complexes I and IV confirmed the increased activity of both complexes (Fig. 1h). Congruent with mitochondrial activation, ATP was also increased in cells treated for 72 h with hydralazine (Fig. 1i). Altogether our data support the hypothesis that hydralazine mediates improvement in mitochondrial function and promotes mitochondrial biogenesis.

**Hydralazine modulates SIRT1/ SIRT5 to activate mitochondria**. Hydralazine appears to mimic CR as we showed previously[18]. Sirtuins (SIRTs), a family of NAD-dependent deacetylases, are nutrient sensors that mediate the prolongevity and health benefits of CR[20,21]. SIRT1 is one of the critical regulators of mitochondrial biogenesis and activation, functioning through peroxisome proliferator-activated receptor γ coactivator 1α (PGC1A) and the mitochondrial transcription factor A (TFAM), two key transcription factors involved in mitochondrial biogenesis[22]. SIRT5 is another member of the SIRT family that is localized at the mitochondrial matrix and controls mitochondrial function by modulating hundreds of enzymes involved in cell metabolism and bioenergetics[22,23]. To investigate the effect of hydralazine treatment on sirtuins, we measured changes in the expression of all sirtuin genes (SIRT1-7) in cells treated with hydralazine. SIRT5 was the only sirtuin that showed upregulation (Fig. 2a and Supplementary Fig. 1a). We hypothesized that SIRT1 is critical in hydralazine-mediated activation of mitochondria considering that PGC1A, a known SIRT5 transcription

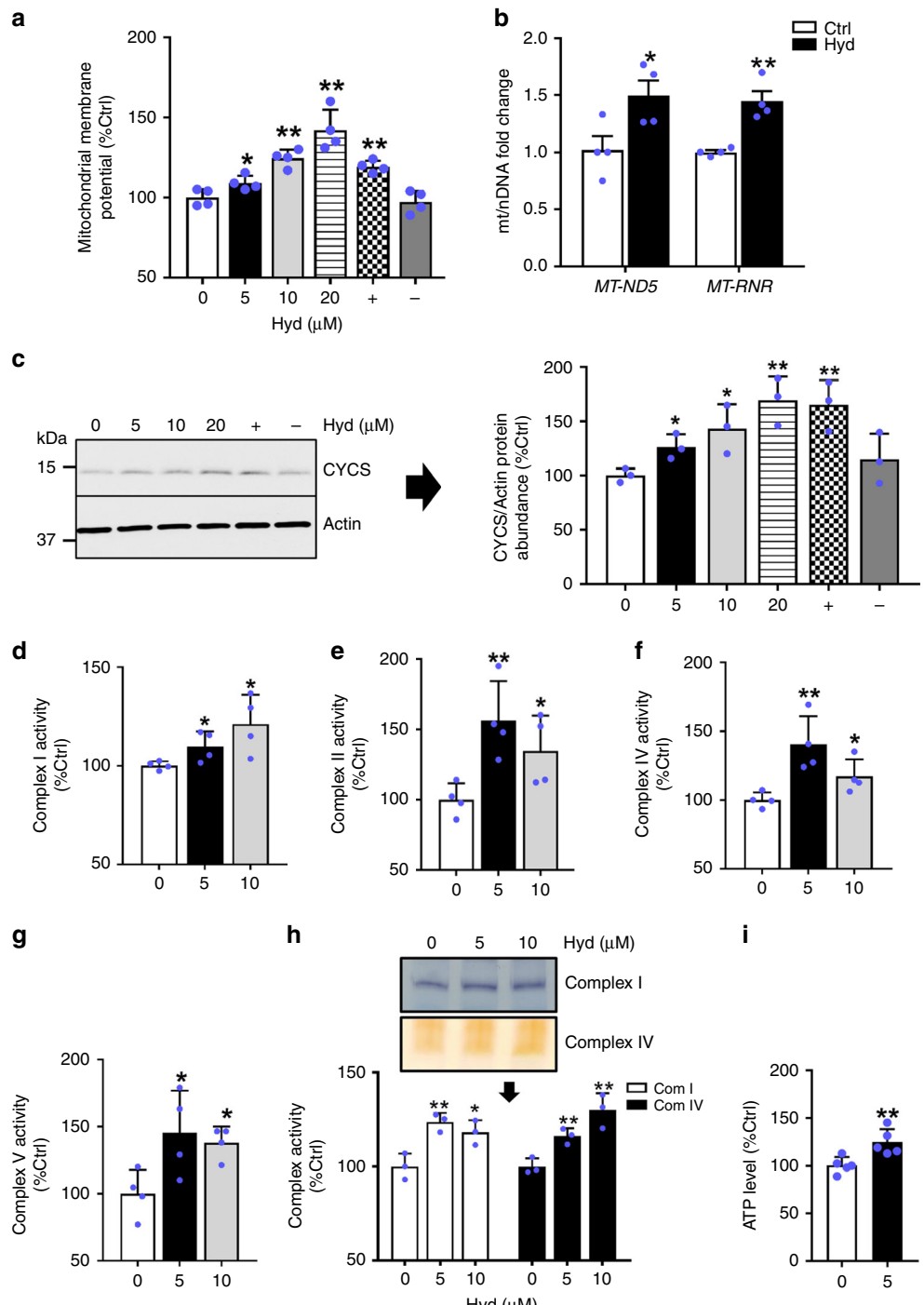

**Fig. 1** Hydralazine improves mitochondrial function. **a** ΔΨm was measured in SH-SY5Y cells treated with different concentrations of hydralazine for 72 h. 10 μM isoniazid or resveratrol were used as negative and positive controls (*n* = 4, mean ± SD). **b** Relative q-PCR analysis of mitochondrial DNA showing an increased mt/nDNA in SH-SY5Y cells treated with hydralazine (5 μM) for 72 h (*n* = 4, mean ± SEM). **c** Hydralazine increases cytochrome c (CYCS) abundance in a dose-dependent manner measured by Western blot analysis in SH-SY5Y cells (*n* = 3, mean ± SD). Spectrophotometric analysis of mitochondrial complexes showing activation of ETC **d** complex I, **e** complex II, **f** complex IV and **g** ATP synthase complex V in C2C12 cells treated with hydralazine for three days (*n* = 4, mean ± SD). **h** In-gel activity assay of the isolated mitochondria showing higher activity for mitochondrial ETC complexes I and IV in C2C12 treated with hydralazine for 72 h (*n* = 3, mean ± SD). **i** The relative amount of ATP in SHY-SY5Y cells treated with 5 μM hydralazine for 72 h (*n* = 5, mean ± SD). *$p \leq 0.05$ and **$p \leq 0.01$, two-tailed Student's *t*-test

activator[22], is tightly regulated by SIRT1. We measured SIRT1 abundance by Western blot analysis and observed a significant increase in cells treated with hydralazine (Fig. 2b). Western blot analysis confirmed the increased abundance of both SIRT5 and TFAM in the treated cells as well (Fig. 2b). We also measured the

enzymatic activity of SIRT1 by tracking the fluorescence signal emitted by a peptide substrate upon deacetylation and observed a higher activity in C2C12 cell extracts treated for 48 h with hydralazine (Fig. 2c). We measured the effect of hydralazine-induced SIRT1 activation on PGC1A deacetylation. PGC1A was

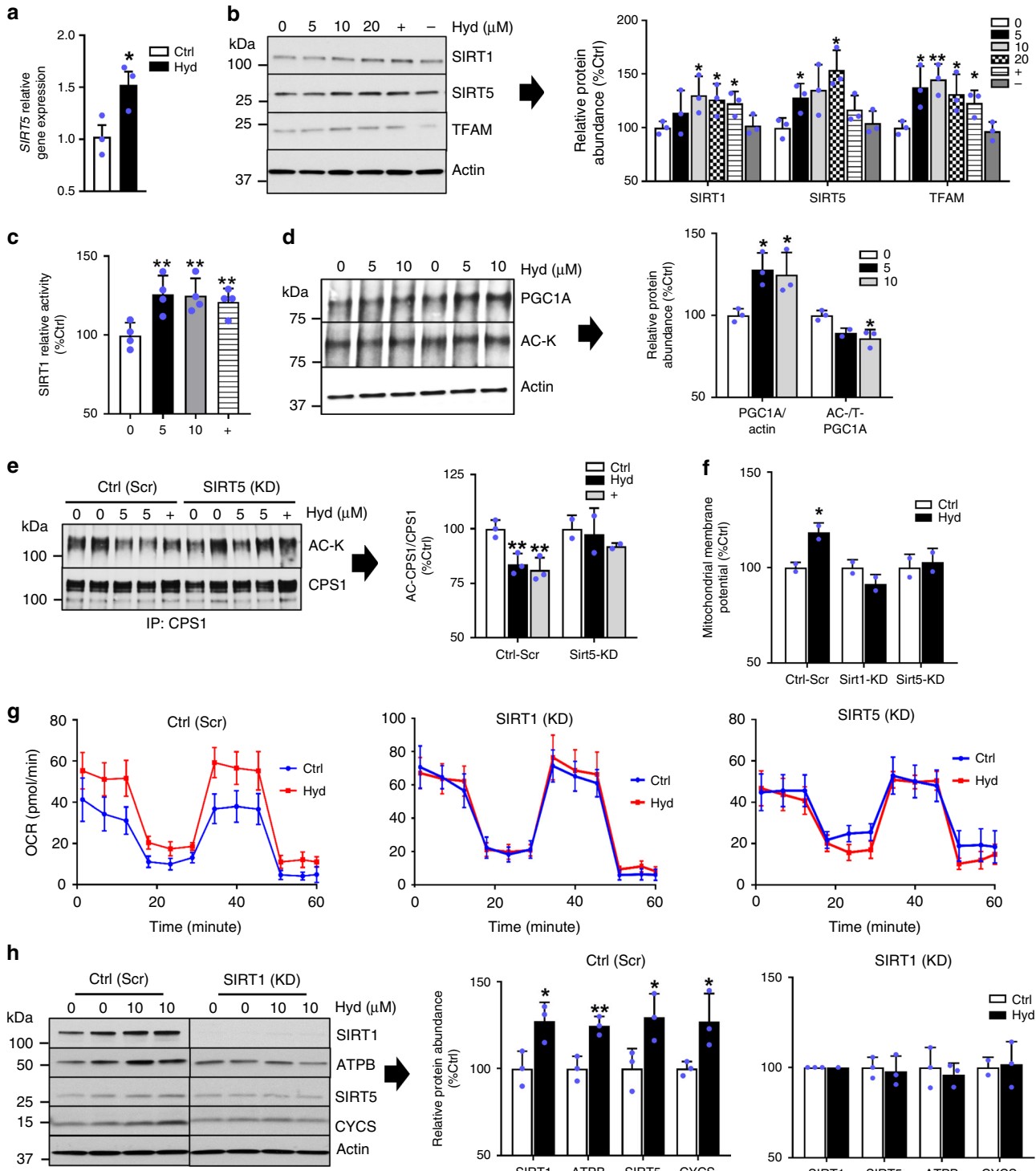

**Fig. 2** Hydralazine activates SIRT1 and SIRT5 to promote mitochondrial function. **a** Hydralazine treatment (10 μM) increases the expression of *SIRT5* mRNA in SH-SY5Y cells measured by qPCR using actin as an internal control (*n* = 3, mean ± SEM). **b** Hydralazine increases protein abundance of SIRT1, SIRT5, and TFAM demonstrated by Western blot analysis. Cells were treated with different doses of hydralazine or 20 μM of resveratrol or isoniazid as positive and negative controls for 24 h (*n* = 3, mean ± SD). **c** SIRT1 enzymatic activity was measured by monitoring the fluorescence emission of an acetylated peptide substrate in C2C12 cells treated with hydralazine for 48 h (*n* = 4, mean ± SD). **d** PGC1A immunoprecipitation followed by Western blot analysis showing activation of PGC1A in hydralazine treated SH-SY5Y cells (*n* = 3, mean ± SD). **e** Western blot analysis demonstrates a SIRT5-dependent increase in deacetylation of CPS1 with hydralazine treatment. 5 μM resveratrol used as a positive control (*n* = 2–3, mean ± SD). **f** Flow cytometric quantification of ΔΨm indicates activation of mitochondria in a SIRT1 and SIRT5-dependent manner (*n* = 2, mean ± SD). **g** Seahorse XF assay demonstrates a SIRT1 and SIRT5-dependent increase in oxygen consumption rate in SH-SY5 cells treated with hydralazine (5 μM, 3 days) (*n* = 3, mean ± SEM). **h** Western blot analysis demonstrating the effects of hydralazine-induced SIRT1 activation on the abundances of SIRT5 and mitochondrial markers in C2C12 cells treated for 48 h (*n* = 3, mean ± SD). *$p \leq 0.05$ and **$p \leq 0.01$, two-tailed Student's *t*-test

immunoprecipitated from cells treated with hydralazine followed by Western blot analysis using an anti-acetylated-lysine antibody. An increase in total PGC1A and a moderate reduction in the acetylated form of the enzyme was observed (Fig. 2d). We also measured SIRT5 activity which is directly linked to mitochondrial function and metabolism. Carbamoyl phosphate synthetase 1 (CPS1), located in the mitochondrial matrix, is the rate-limiting enzyme of the urea cycle and one of the substrates of SIRT5[24]. Reduced acetylation of CPS1, measured by immunoprecipitation (IP) and Western blot analysis, confirmed the increased activity of SIRT5 with hydralazine treatment (Fig. 2e).

To determine the direct role of the SIRT1/SIRT5 signaling pathway in the mitochondrial function, we knocked down SIRT1 and SIRT5 in SH-SY5Y cells (Supplementary Fig. 1b–c) and measured Δψm in the control and knockdown cells treated with hydralazine (10 μM) for three days. Hydralazine treatment was accompanied by a significant increase in Δψm in cells transfected with control shRNA which was abrogated in the absence of SIRT1 and SIRT5 (Fig. 2f). In parallel, we used two known inhibitors of SIRT1, nicotinamide (NAM), a byproduct of SIRT1 enzymatic activity, and sirtinol. Δψm did not increase with hydralazine treatment (20 μM for 72 h) in the presence of both SIRT1 inhibitors, indicating that SIRT1 is essential for hydralazine-induced mitochondrial activation (Supplementary Fig. 1d). We also observed a SIRT1 and SIRT5-dependent increase in respiration with hydralazine treatment as measured by Seahorse XF (Fig. 2g). Our data showed a SIRT1-dependent increase in protein abundance of SIRT5 and a few other mitochondrial markers including ATP synthase beta subunit (ATPB), and CYCS with hydralazine treatment indicating an upstream role for SIRT1 in promoting SIRT5 expression and mitochondrial function (Fig. 2h).

Hydralazine is known to inhibit prolyl hydroxylase domain (PHD) enzymes and activates the hypoxia-inducible factor 1 alpha (HIF1A) pathway[25]. However, we did not observe any link between HIF1A and activation of mitochondria with hydralazine (Supplementary Fig. 1e). A connection between HIF1A and protective effects of hydralazine against rotenone toxicity was not observed in C. elegans either (Supplementary Fig. 1f).

**Mitochondrial metabolism and hydralazine longevity benefit.**
To further investigate the biological effects of hydralazine on mitochondrial metabolism, we performed a global metabolomics analysis. We quantified 140 known metabolites across three biological replicates extracted from control and hydralazine treated (4 days, 100 μM) wild-type C. elegans (Supplementary Data 1). The post-quality-controlled data were used for multivariate principal component analysis (PCA). The applied analysis discriminated the data from control and hydralazine treated animals demonstrating distinct metabolic profiles (Fig. 3a). A heat map of the top 50 affected metabolites revealed a general change in bioenergetic intermediates (Fig. 3b). Metabolic pathway analysis of significantly up or downregulated metabolites (p < 0.05) identified the TCA cycle as the top enriched pathway (Fig. 3c). Elevated levels of lactate and pyruvate as well as perturbed NADH/NAD+ and lactate/pyruvate ratios are biomarkers of mitochondrial dysfunction[26]. Our data did not detect any change in these biomarkers. Instead, we observed a significant increase in NAD and ATP with hydralazine treatment (Fig. 3d). An overall elevation in TCA cycle intermediates representing a higher flux and activation of mitochondria was also observed (Fig. 3e). We measured the rate of whole-body oxygen consumption in worms grown under the same conditions. Congruently a higher rate of respiration was observed in hydralazine-treated worms (Fig. 3f).

Next, we investigated the link between increased mitochondrial activity and prolongevity effects of hydralazine using antimycin A, an inhibitor of mitochondrial complex III, and two different C. elegans strains carrying mutations in mitochondrial ETC complexes I and II (gas-1 and mev-1). Lifespan extension was not observed in C. elegans grown in medium containing low concentrations of antimycin A (Fig. 3g). Similarly, neither of the mutant C. elegans with dysfunctional mitochondria experienced lifespan extension with hydralazine treatment (Fig. 3h, Supplementary Fig. 2). Animals lacking functional SIR-2.1 (SIRT1 ortholog in C. elegans), which is essential for hydralazine-induced mitochondrial activation, also did not experience lifespan extension (Fig. 3h). No change was observed in Δψm in the mutant worms (sir-2.1, gas-1, and mev-1) that did not experience lifespan extension with hydralazine treatment (Fig. 3i). As expected, C. elegans strain skn-1 (zu135) with loss of function mutation in all SKN-1 isoforms that did not experience hydralazine-mediated lifespan extension[18], also did not show mitochondrial activation with hydralazine (Fig. 3i). Congruent with the essential role of ASI neuronal SKN-1 in the extension of lifespan[18], the presence of neuronal SKN-1 in skn-1 (zu67) strain recapitulated the beneficial effects of hydralazine on mitochondrial function (Fig. 3i). Altogether these data suggest a correlation between activation of mitochondria and extension of lifespan in C. elegans mediated by hydralazine.

**Protective benefits of hydralazine on mitochondrial function.**
We measured mitochondrial health and function to determine the efficacy of hydralazine administration in protecting against different cellular stressors which are usually associated with the pathophysiology of age-related disorders.

A high glucose regimen is associated with mitochondrial dysfunction, reduced respiration, and decreased lifespan in C. elegans[27,28]. Wild-type nematodes grown in medium containing a high concentration of glucose (50 mM) showed a reduced median lifespan (from ~17 to ~14 days). Hydralazine treatment protected the nematodes from glucose toxicity and increased their median lifespan (from ~14 to ~16 days). (Fig. 4a, Supplementary Table 1). We also monitored the locomotor performance of worms on a high glucose diet with and without hydralazine treatment. Hydralazine significantly decelerated the deterioration of locomotor performance in 10-days-old animals on a high glucose diet (Fig. 4b). Hydralazine reversed Δψm declined as a result of glucose toxicity (Fig. 4c). Seahorse analysis of worms on a high glucose diet revealed a decrease in respiration which was restored with hydralazine (Fig. 4d). Metabolomics analysis of the TCA cycle and bioenergetics intermediates revealed a disturbed mitochondrial metabolism induced by a high glucose diet (Supplementary Fig. 3a). Hydralazine reverted the glucose-induced perturbation of the metabolism and overall health of the mitochondria to those of the control animals (Supplementary Fig. 3a). Mitochondria are highly dynamic interconnected organelles requiring fusion and fission for their viability and response to changes in cellular bioenergetics[29–31]. While an elongated mitochondrial morphology is associated with healthy efficient mitochondria (i.e., higher ATP production and lower ROS generation), a fragmented morphology is linked to mitochondrial dysfunction[30]. To determine the detrimental effects of high glucose on mitochondrial morphology, we used transgenic C. elegans overexpressing mitochondrial matrix-targeted GFP in body-wall muscle cells [Pmyo-3::GFP(mit)][32]. The microscopic images revealed a fragmented mitochondrial morphology in eight days-old nematodes grown in high glucose. However, hydralazine-treated groups showed a less fragmented, more tubular morphology,

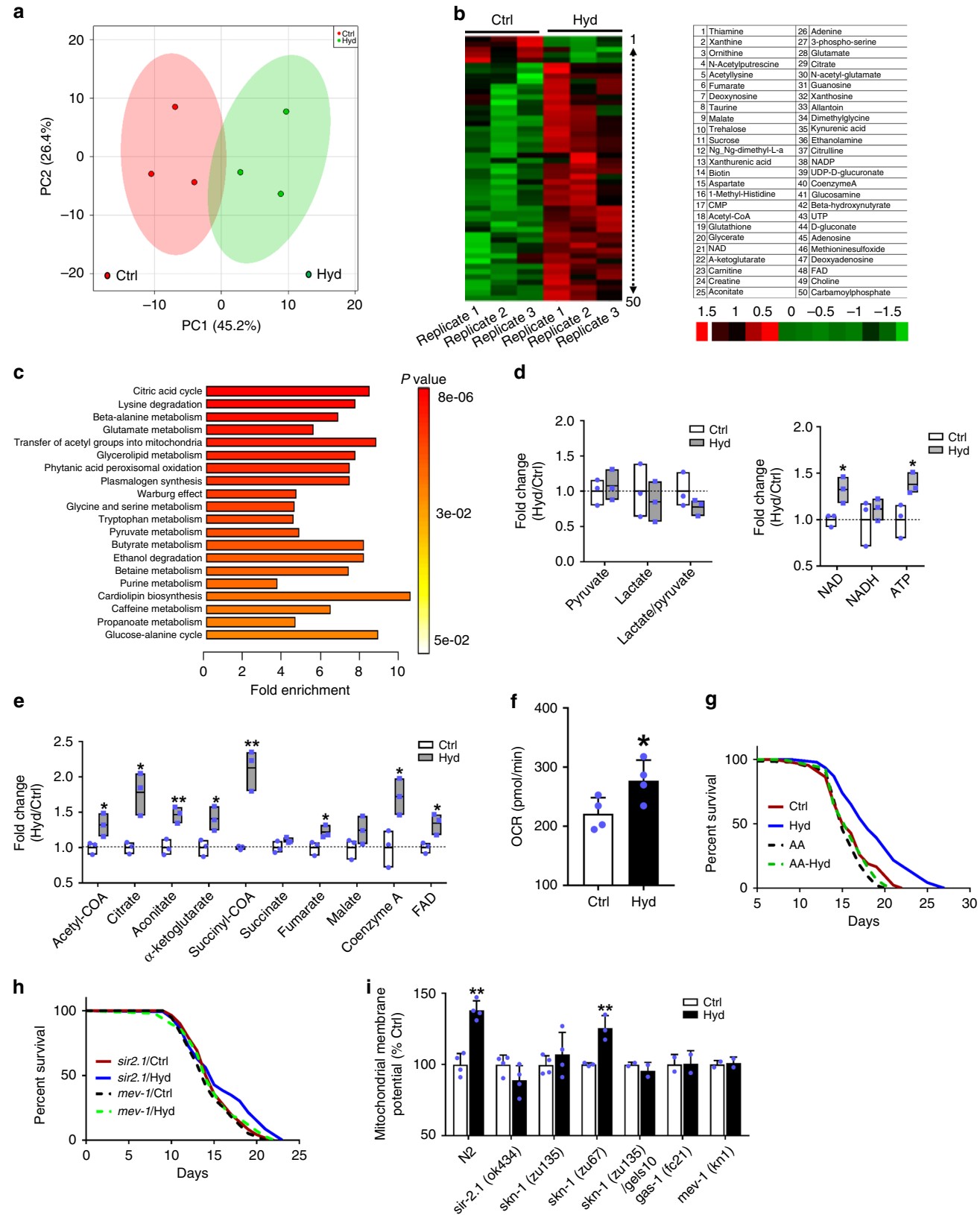

demonstrating hydralazine's protective effect against glucose-induced mitochondrial fragmentation (Fig. 4e). These data demonstrated the detrimental effect of high glucose diet on *C. elegans* mitochondrial metabolism and the ability of hydralazine to effectively reverse the negative impact of high glucose.

A pro-aggregant transgenic *C. elegans* strain which pan-neuronally expresses a rab-3 promoter-driven, highly amyloido-genic, mutated F3ΔK280 fragment of human tau, was used as an in vivo tauopathy model. An anti-aggregant F3ΔK280-PP transgenic strain was used as control[33]. We had previously shown

**Fig. 3** Hydralazine alters mitochondrial metabolism and activity to extend *C. elegans* lifespan. **a** Principal-component analysis of metabolomics data discriminates hydralazine treated wild-type *C. elegans* (100 μM, 4 days) from control. **b** Heatmap plot showing the top 50 metabolites affected by hydralazine treatment. Downregulated and upregulated metabolites are shown in green and red, respectively. **c** Metabolite Set Enrichment Analysis reveals mitochondrial citric acid cycle as the most upregulated pathway in hydralazine treated *C. elegans*. **d, e** Mass spectral analysis of individual metabolites representing bioenergetics and TCA cycle intermediate ($n = 3$). **f** Seahorse XF assay shows an increased rate of $O_2$ consumption in wild-type *C. elegans* treated with hydralazine (100 μM, 4 days) ($n = 4$). **g** Prolongevity effects of hydralazine are inhibited in wild-type *C. elegans* grown in the presence of mitochondrial complex III blocker, antimycin A. **h** Prolongevity effects of hydralazine is inhibited in *C. elegans* lacking an active SIR-2.1 or having an impaired mitochondrial ETC in complex II (*mev-1* mutants). See Table S1 for lifespan statistics. **i** Quantification of TMRE fluorescence signal demonstrates a link between longevity benefits of hydralazine and activation of mitochondria ($n = 4$ for N2, *sir-2.1* and *skn-1*(zu135) groups, $n = 3$ for *skn-1* (zu69), and $n = 2$ for the rest of the measurements). *$p \leq 0.05$ and **$p \leq 0.01$, two-tailed Student's *t*-test, mean ± SD

the beneficial effects of hydralazine on the survival and health of this tauopathy *C. elegans*[18]. Here we investigated the overall benefit of hydralazine on mitochondrial function in this model by measuring mitochondrial membrane potential. A significant reduction in $\Delta\Psi m$ was observed in the pro-aggregant strain compared to the anti-aggregant strain. $\Delta\Psi m$ was significantly increased in animals treated with hydralazine (Fig. 4f). HEK293 cells overexpressing tau residues 244 to 372 with mutations of P301L and V337M seeded with recombinant tau fibrils to indefinitely propagating tau aggregates (aggregate-positive cells, AP), were used as an in vitro model for tauopathy. HEK293 cells not seeded with recombinant tau fibrils were used as control (aggregate-negative cells, AN)[34]. $\Delta\Psi m$ compromised by tau aggregates was restored with hydralazine in AP cells (Fig. 4g). To test the role of sirtuins in the hydralazine-mediated recovery of mitochondrial function in tauopathy cell models, we treated SIRT1 and SIRT5 knockdown AP cells with hydralazine. Cell viability and $\Delta\Psi m$ restoration in AP cells were abrogated in the absence of either SIRT1 or SIRT5 (Fig. 4g, Supplementary Fig. 3b). Western blot analysis showed that the abundance of both SIRT5 and TFAM was decreased by tau aggregation, confirming the detrimental effects of tau aggregates on mitochondrial integrity. However, hydralazine treatment restored both SIRT5 and TFAM proteins to the control levels (Supplementary Fig. 3c). The overall cell bioenergetic state and mitochondrial function measured by $\Delta\Psi m$, as well as key regulatory elements of mitochondrial function, such as SIRT5 and TFAM, were adversely affected by tau aggregates and were restored by hydralazine.

Rotenone is a mitochondrial complex I inhibitor that induces mitochondrial dysfunction and symptoms similar to Parkinson's disease[35]. We recently reported the protective effect of hydralazine against rotenone with a mechanism which is partially mediated through NRF2/SKN-1 pathway[18]. In addition to NRF2/SKN-1, our proteomics analysis showed upregulation of mitochondrial proteins in hydralazine treated nematodes (Supplementary Fig. 3d). In accordance with our proteomics data, we investigated the possible role of two key bioenergetic regulators, SIRT1 and AMP-activated protein kinase (AMPK). AMPK is one of the cell's key bioenergetics regulators involved in the aging process via multiple pathways, including NRF2/SKN-1, SIRT1, and mitochondrial activation[36,37]. To deconvolute the protective role of SIRT1 from AMPK against rotenone, we used single and double mutant strains of SIR-2.1 and AAK-2 (AMPK orthologue in *C. elegans*). We observed protection against a lethal dose of rotenone (50 μM) only in wild-type and AAK-2 mutant worms indicating that hydralazine-mediated protection is SIR-2.1-dependent (Fig. 4h). Collectively these data support our hypothesis regarding the activation of the SIRT1 and highlight its pivotal role in beneficial effects of hydralazine in mitochondrial health under stress conditions.

**PKA as the direct target of hydralazine's longevity benefit**. Improvement of mitochondrial function by SIRT1/SIR-2.1 and

activation of NRF2/SKN-1 pathway are essential for the pro-longevity and health benefits of hydralazine in vitro and in vivo. However, the direct binding target of hydralazine has remained unknown. To identify the direct molecular target of hydralazine, we utilized an unbiased target identification technique (i.e., Thermal proteome profiling (TPP)) using *C. elegans* as the model organism[38]. Melting curves of 1,802 soluble proteins from control and hydralazine treated living animals across ten different temperatures were acquired using isobaric mass tagging (TMT10-plex) in combination with high-resolution mass spectrometry. The catalytic subunit of PKA complex was stabilized in the presence of hydralazine identifying it as hydralazine's direct target while the regulatory subunit was destabilized, an indication of indirect interaction with hydralazine (Fig. 5a). It has been previously shown that PKA complex will dissociate in the presence of a direct binding ligand to either subunit yielding an opposite shift in thermal stability curves for the two subunits[39]. Western blot analysis confirmed a dose-dependent stabilization of the catalytic subunit by hydralazine in HEK293 cells (Fig. 5b). Our in silico docking simulation also supports the direct binding of hydralazine to a pocket far from the active site of the catalytic subunit of PKA with a binding affinity of −6.2 (kcal/mol) (Fig. 5c, Supplementary Fig. 4a). Modeling indicated formation of two hydrogen bonds between the hydrazide group of hydralazine and leucine 50 and glutamic acid 128 of the PKA catalytic subunit, highly evolutionary conserved residues (Fig. 5c). We also measured the relative amount of cAMP in hydralazine treated cells and ruled out a cAMP-dependent mechanism for dissociation of the PKA complex (Supplementary Fig. 4b). Our data collectively support the conclusion that hydralazine binds directly to the PKA catalytic subunit resulting in dissociation from the PKA regulatory subunit.

Dissociation of the regulatory subunit from the catalytic subunit is a prerequisite for PKA pathway activation[12]. We assayed kinase activity in the presence and absence of hydralazine with radiolabeled ATP in crude cell extracts and observed a hydralazine-stimulated increase in phosphorylation of histone protein substrates in the wild-type cells. Phosphorylation of protein substrates was abrogated in PKAcat-KO cells (Fig. 5d). We further confirmed these results by measuring phosphorylation of cAMP response element-binding protein (CREB), a well-established downstream target of PKA (Fig. 5e).

PKA directly phosphorylates and consequently activates SIRT1 via a NAD-independent mechanism[13]. We measured the activity of SIRT1 in the presence and absence of hydralazine and observed a PKA-dependent increase in SIRT1 relative activity (Fig. 5f). A significant PKA-dependent increase in protein abundance of SIRT1 was also observed with hydralazine treatment (Supplementary Fig. 4d). The elevation of mitochondrial membrane potential in both *C. elegans* and HEK293 cells treated with hydralazine was also PKA-dependent (Fig. 5g, Supplementary Fig. 4e). Protection against rotenone toxicity (measured as percent

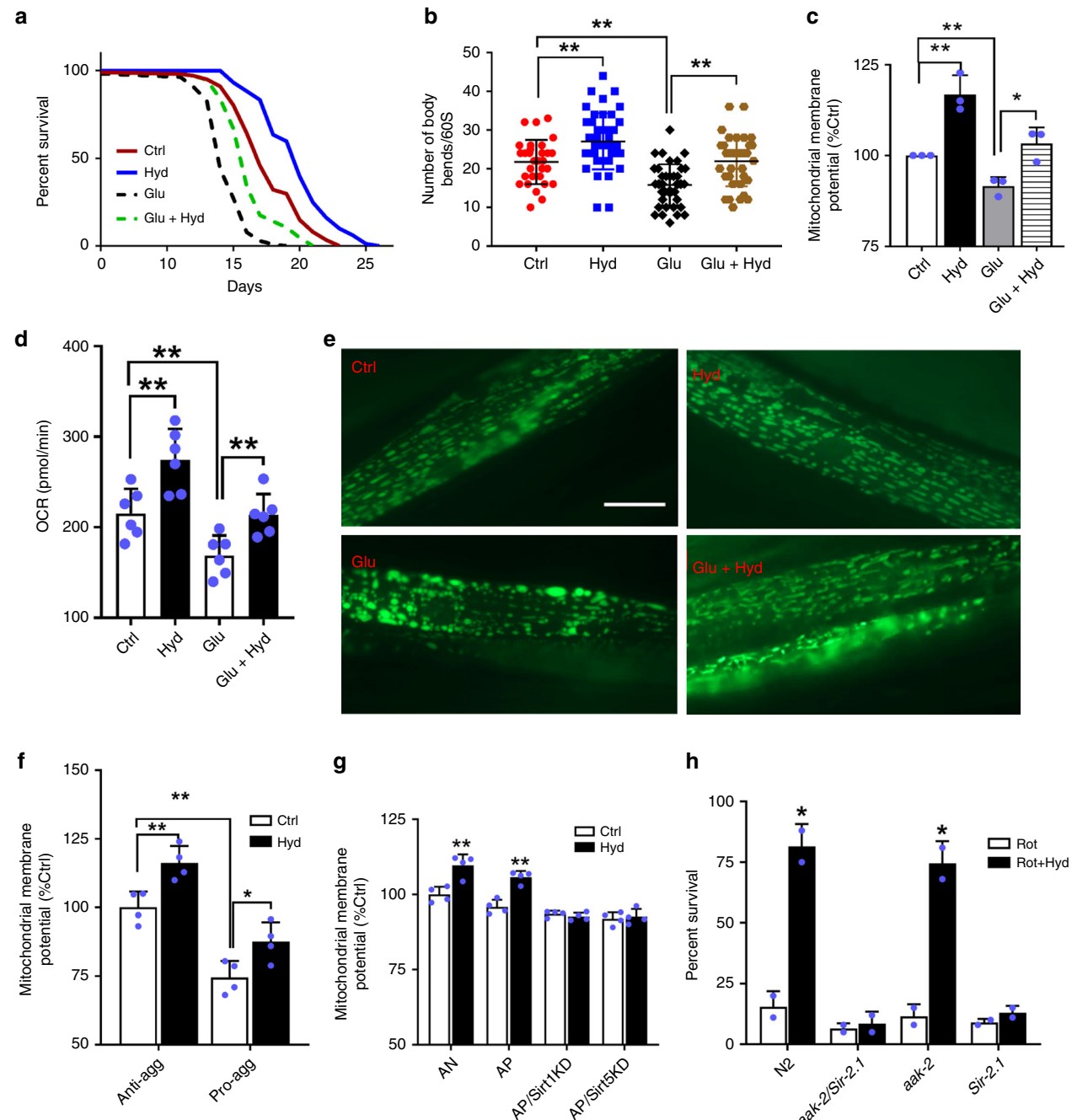

**Fig. 4** Hydralazine protects mitochondrial function under different stress conditions. **a** Hydralazine treatment restores the lifespan of wild-type nematodes grown under high glucose conditions. See Table S1 for lifespan statistics. **b** Motility assay demonstrates that reduced locomotor performance in 10 days old *C. elegans* grown under high glucose condition is restored with hydralazine treatment (from left to right $N = 30, 43, 40, 36$). **c** Measurement of ΔΨm shows that hydralazine treatment protects mitochondrial function in *C. elegans* grown under high glucose diet ($n = 3$). **d** Seahorse XF assay reveals compromised respiration induced by a high glucose diet. Compromised respiration is rescued in the presence of hydralazine in 10 days old nematodes ($n = 6$). **e** Fluorescence microscopy photomicrographs of *C. elegans* expressing mtGFP reveals protection of mitochondrial morphology under hydralazine treatment in the nematodes grown under high glucose condition for 8 days. Scale bar = 20 µM. **f** Measurement of ΔΨm shows that hydralazine treatment (100 µM, 4 days) restores the compromised mitochondrial function in *C. elegans* model of tauopathy expressing pro-aggregant tau fragment in the nervous system (*byIs161*) ($n = 4$). **g** Quantification of TMRE fluorescence signal in AP cells treated with hydralazine (10 µM, 3 days) demonstrates a SIRT1 and SIRT5 dependent increase in mitochondrial ΔΨ ($n = 4$). **h** The percentage of wild-type and mutant *C. elegans* survived under rotenone toxicity shows a sirtuin-dependent (not AMPK) mechanism of action for hydralazine (from left to right $N = 85, 75, 78, 130$, two independent experiments). *$p ≤ 0.05$ and **$p ≤ 0.01$, two-tailed Student's *t*-test, mean ± SD

survival) in animals treated with hydralazine was abrogated in *kin-1* RNAi fed animals which confirmed the critical role of PKA orthologue in hydralazine-mediated protection in *C. elegans* (Fig. 5h). The prolongevity benefit of hydralazine was also abrogated in KIN-1 knockdown animals (Fig. 5i).

We recently showed that hydralazine activates the NRF2 pathway with a mechanism that does not involve direct dissociation of NRF2 from KEAP1 or a hormetic response[18]. Western blot analysis of NRF2 and one of the downstream enzymes of ARE, heme oxygenase-1 (HMOX1) showed a PKA-

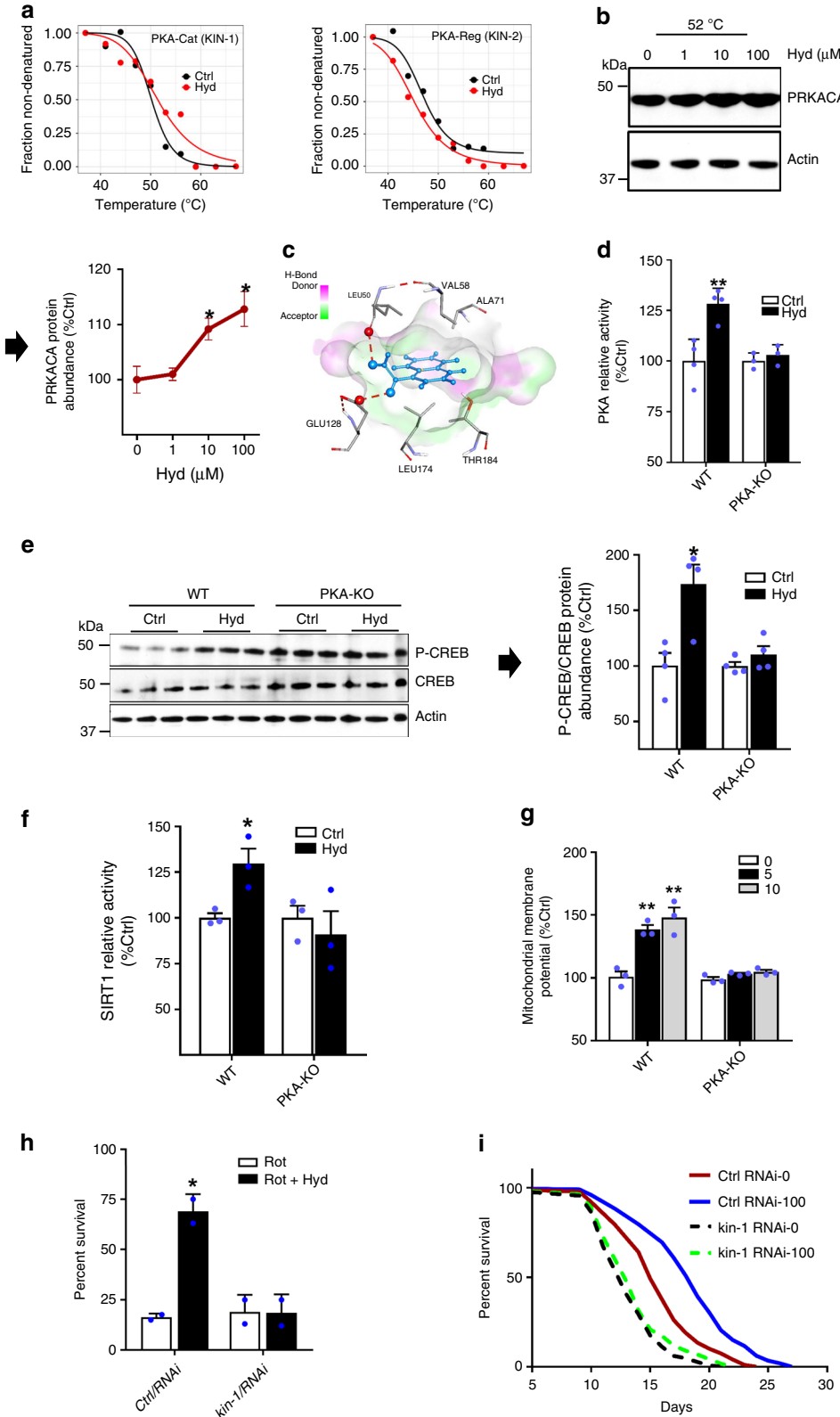

dependent upregulation in cells treated with hydralazine (Supplementary Fig. 4d). Collectively, these observations support the role of PKA in hydralazine-induced NRF2 activation.

**High concentrations of hydralazine prohibit cell respiration**. In this study, most of the parameters related to mitochondrial activity were measured after two or more days of hydralazine

treatments. We measured a few markers of mitochondrial function in order to understand how mitochondria respond to hydralazine at earlier times. Oxygen consumption rates (OCR) in C2C12 cells treated with different concentrations of hydralazine for four hours were measured by Seahorse. Surprisingly, a dose-dependent reduction in the rate of oxygen consumption was observed with hydralazine (Fig. 6a). The rate of oxygen

**Fig. 5** Hydralazine binds and activates PKA. **a** Melting curve of PKA measured by Thermal proteome profiling in *C. elegans* demonstrates that the catalytic subunit of PKA complex is stabilized in the presence of hydralazine whereas the regulatory subunit is destabilized. **b** Hydralazine stabilizes the catalytic subunit of PKA in a dose-dependent manner in HEK293 cells heated for 3.5 min ($n = 3$ for 1 μM and $n = 4$ for the rest of the measurements, mean ± SEM). **c** In silico simulation and molecular dynamics analysis of PKA showing the interaction of hydralazine with the catalytic subunit of PKA. **d** Histone phosphorylation assay demonstrating a higher activity of PKA in HEK293 cells treated with hydralazine (10 μM) for 2 h ($n = 3$ for PKA-KO and $n = 4$ for the WT, mean ± SD). **e** Western blot analysis demonstrating PKA-dependent phosphorylation of CREB in HEK293 cells treated with 10 μM hydralazine for 4 h ($n = 4$, mean ± SEM). **f** SIRT1 activity assay demonstrating a PKA-dependent elevation in SIRT1 activity in HEK293 cells treated with hydralazine (10 μM) for 48 h ($n = 3$, mean ± SEM). **g** Quantification of TMRE signal by microplate reader demonstrating a PKA-dependent elevation in Δ$\Psi$m in HEK293 cells treated with 0, 5, and 10 μM of hydralazine for 48 h ($n = 3$, mean ± SEM). **h** Survival rate indicating KIN-1-dependent rotenone protection in *C. elegans* treated with hydralazine (100 μM) for three days ($N = 150$, two independent experiments, mean ± SD). **i** Lifespan assay demonstrates a PKA-dependent lifespan extension in *C. elegans* treated with hydralazine. See Table S1 for lifespan statistics. *$p \leq 0.05$ and **$p \leq 0.01$, two-tailed Student's *t*-test

consumption was low in cells treated with higher concentrations of hydralazine for longer exposure times (i.e., 72 h) and only was increased in cells treated with lower concentrations of hydralazine (below 5 μM) (Supplementary Fig. 5a). One explanation for these data is hydralazine-induced activation of stress response pathways such as the mitochondrial unfolded protein response (UPR$^{mt}$). To test this hypothesis, we treated C2C12 cells with 20 μM hydralazine (a concentration that significantly impairs $O_2$ consumption according to our previous experiments) and quantified Δ$\Psi$m from 4 h to 72 h. Contradictory to oxygen consumption data, a time-dependent increase in Δ$\Psi$m with the maximum effect at 72 h was observed (Fig. 6b). We also measured ETC complex IV (cytochrome c oxidase) activity and did not observe a hydralazine-induced impairment of this complex. ETC complex IV is the last enzymatic complex in the respiratory electron transport chain of mitochondria where oxygen is converted to water, a possible inhibitory site of respiration. Congruent with mitochondrial Δ$\Psi$m elevation, a time-dependent activation of ETC complex IV was observed (Fig. 6c).

We also monitored UPR$^{mt}$ induction in transgenic *C. elegans* expressing a *hsp-6::GFP* reporter. A synchronized population of *C. elegans* from larval and young adult stages was exposed to a range of hydralazine doses, up to 1 mM from hours to days. The microscopic data did not show any increase in the fluorescence signal indicative of activation of UPR$^{mt}$ by hydralazine (Supplementary Fig. 5b). These data ruled out the possibility of hydralazine-induced mitochondrial dysfunction as the main mode of action for this phenotype and suggested an off-target effect or pleiotropic effects of PKA at higher doses of hydralazine.

Mitochondrial health parameter measurements appear to vary in a dose-dependent manner which may affect the physiological outcome and health benefits of hydralazine therapy (Fig. 6d).

Our data collectively demonstrate that hydralazine activates PKA, independent of cAMP to trigger activation of different pathways including SIRT1 and NRF2, modulating mitochondrial function to provide health and prolongevity benefits (Fig. 6e).

## Discussion

Hydralazine was approved as a vasodilator by FDA in 1953 and because of its effectiveness and benign side effects is still in clinical use. Despite repeated attempts at identifying underlying mechanisms, its direct target has remained unknown. Recently our group and several others have reported prolongevity benefits of hydralazine in multiple organisms including yeast, rotifer and *C. elegans*[15,18,40]. We showed that hydralazine-mediated life and healthspan extension and protection against rotenone toxicity in *C. elegans* are the results of NRF2/SKN-1 pathway activation[18]. While SKN-1 activation explained the prolongevity effects of hydralazine in *C. elegans*, it did not fully explain rotenone-induced stress resistance and improved locomotor performance under stress condition[18]. Our global comparative quantitative proteomics screen revealed upregulation of mitochondrial

proteins (Supplementary Fig. 3d). Therefore, we searched for a possible secondary protective mechanism(s) that link mitochondrial function to prolongevity and stress protection benefits of hydralazine. We demonstrate that hydralazine improves mitochondrial function in a SIRT1/SIRT5-dependent manner and that the activation of mitochondria is necessary for the extension of lifespan. In order to identify the target of hydralazine, we performed an unbiased proteomics screen (TPP) in *C. elegans* and identified the catalytic subunit of PKA as a target for hydralazine. We demonstrated that activation of PKA is necessary for upregulation of SIRT1 and NRF2 signaling pathways and consequently, extension of lifespan and protection against rotenone toxicity (Figs. 4, 5h).

The direct link between PKA and SIRT1 has been demonstrated[13]. Hydralazine-induced NRF2 activation can also be explained via a PKA-dependent mechanism (Supplementary Fig. 4e). It has been shown that cAMP/PKA and SIRT1 are upstream regulators of the NRF2-ARE pathway in fasting mice and human hepatocytes[41]. Although our preliminary data suggest a possible role of SIRT1 as an upstream activator of NRF2 (Supplementary Fig. 5c), we cannot rule out NRF2 activation via other PKA-dependent regulatory mechanisms such as CREB or GSK3B[42,43]. Further studies in mammalian cells and *C. elegans* are required to unravel the exact mechanism by which PKA mediates activation of SIRT1 and NRF2. Our data disagree with a study conducted by Gang and Robinson (1995) who found an inhibitory effect of hydralazine on PKA (>30 μM)[44]. This discrepancy can be explained by the fact that the authors studied the effect of hydralazine only on the isolated catalytic subunit of PKA rather than effects in cells.

Actions of PKA signaling in cardiovascular function and vasorelaxation are well known[45,46]. PKA also regulates the activity of sarcoplasmic reticulum $Ca^{2+}$-ATPase (SERCA2) in reuptake of $Ca^{2+}$ into the sarcoplasmic reticulum, which is implicated in heart failure[12]. Activation of SERCA2a has been proposed as the mechanism by which hydralazine promotes myocyte contractility[47]. Although we cannot rule out other mechanism(s), activation of PKA by hydralazine may be the underlying mechanism explaining the effectiveness of the drug in treating hypertension and heart failure.

We also investigated the impact of mitochondrial activation on the prolongevity effect of hydralazine in *C. elegans*. The impact of mitochondrial health and activation on aging is complex and controversial[48]. A mild reduction in mitochondrial activity has been shown to extend lifespan in *C. elegans* and killifish[49]. On the other hand, increasing mitochondrial function and respiration has been shown to mediate longevity benefits of CR[50]. We demonstrated that inhibition of mitochondrial activity, either chemically or genetically, blocks the prolongevity effects of hydralazine in *C. elegans* (Fig. 3g–i). In agreement with our previously published lifespan data[18], the presence of neuronal ASI but not intestinal SKN-1 is necessary for hydralazine-mediated

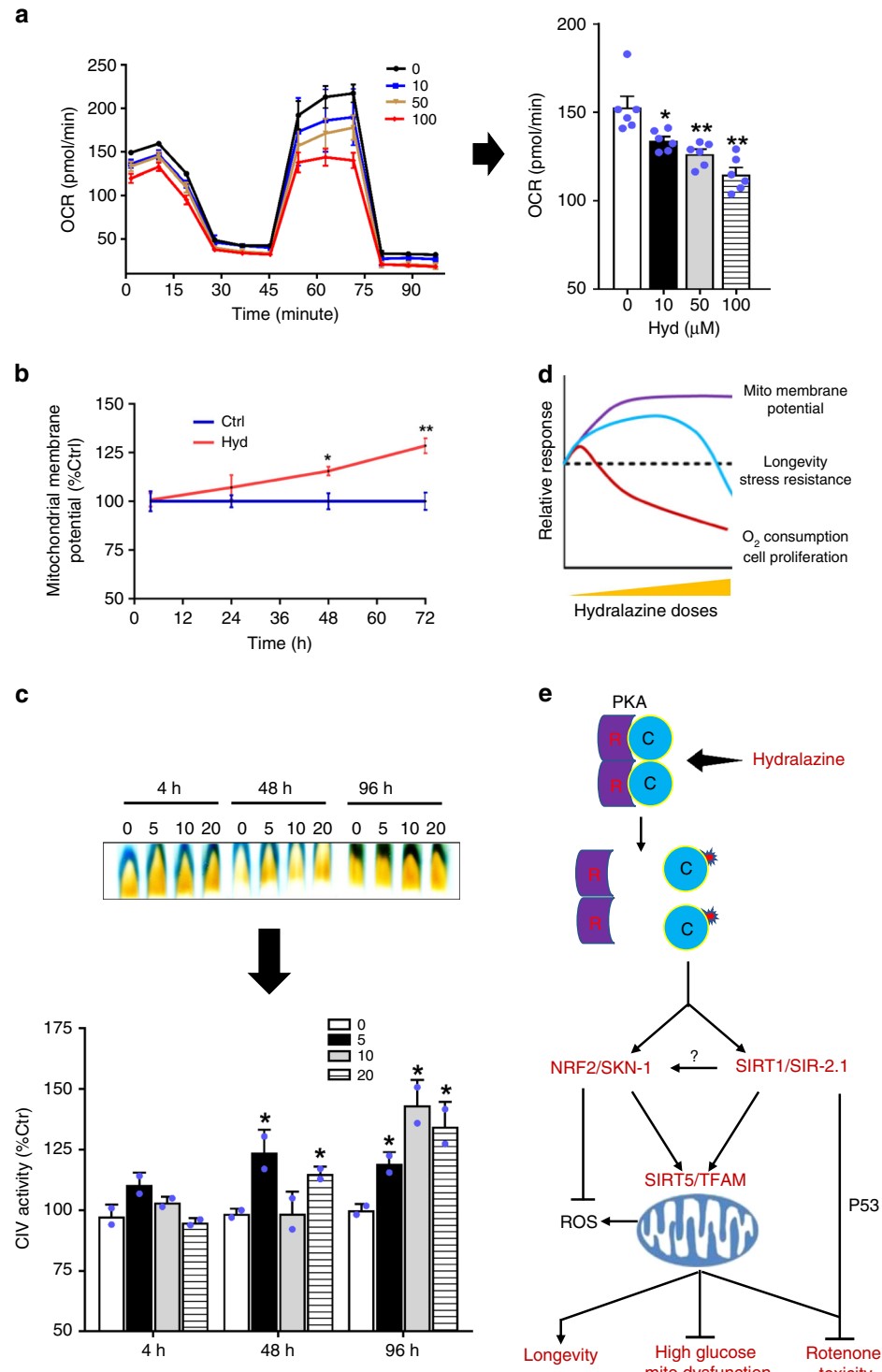

**Fig. 6** Time and dose-dependent effect of hydralazine on mitochondrial respiration and function. **a** Seahorse XF assay demonstrates inhibition of oxygen consumption in response to higher doses (above 10 μM) of hydralazine in C2C12 cells treated for 4 h ($n = 6$, mean ± SEM). **b** Time course measurement of mitochondrial membrane potential reveals a time-dependent activation of mitochondria in hydralazine-treated (10 μM) C2C12 cells ($n = 3$, mean ± SD). **c** In-gel activity assay of the mitochondria isolated from C2C12 treated cells (4, 48, and 96 h) demonstrates a time-dependent activation of mitochondrial ETC complex IV ($n = 2$, mean ± SD). **d** A summary of the dose-dependent response of hydralazine treatment on mitochondrial activity, respiration, longevity and stress resistance in vitro and in *C. elegans*. **e** Hydralazine proposed mechanism of action vis-à-vis lifespan extension and stress protection. $^*p \leq 0.05$ and $^{**}p \leq 0.01$, two-tailed Student's *t*-test

activation of mitochondria (Fig. 3i). A possible explanation for the complementary effect of intestinal SKN-1 in prolongevity benefit of hydralazine is the critical role that intestinal SKN-1 plays in ROS homeostasis that elevates after mitochondria activation (Fig. 6e)[51]. The activation of intestinal SKN-1 with hydralazine decreases the ROS generated by activated mitochondria which is known to have an adverse effect on health and lifespan.

Our data also indicate that hydralazine, at higher doses, diminishes respiration in vitro (Fig. 6a), but we did not observe temporary impairment of mitochondrial function at the same doses (Fig. 6b–d). A reduction in oxygen consumption in *C. elegans* was not observed, at least at the concentration that was used for our studies (10× the concentration used for all in vitro studies) (Figs. 3f, 4e). Measuring hydralazine concentrations inside *C. elegans* cells may be of benefit to understand the link between the rate of oxygen consumption and prolongevity effects of hydralazine. The reduced mitochondrial respiration observed at higher hydralazine doses perhaps accounts for its known anti-cancer efficacy and side effects. Notably, the adverse effects of high concentrations of hydralazine (>200 μM) on mitochondrial function and induction of apoptosis have been reported in leukemic T cells[52].

*C. elegans* has been put forward as a suitable model organism to study glucose toxicity resulting from prolonged hyperglycemia in diabetic patients[27,53]. A high-sugar diet has been associated with reduced lifespan in organisms ranging from worms to mammals[53]. Mitochondria, with a central role in metabolism, become dysfunctional during diabetes[54]. High glucose intake is linked to reduced mitochondrial oxygen reserve capacity[55], elevated oxidative stress, and fragmentation[56]. Hydralazine has been shown to enhance the viability of mouse primary neurocyte experiencing glucose-induced vulnerability to oxidative stress by an unknown mechanism[57]. Using *C. elegans* as a model organism, we demonstrated that hydralazine ameliorates glucose-induced mitochondrial dysfunction by recovering mitochondrial metabolism and respiration and by preserving a more dynamic mitochondrial network which results in reversion of the negative consequences and improves lifespan and health (Fig. 4a–e). These data suggest a potential anti-diabetic efficacy for hydralazine.

Although the metabolic benefits of hydralazine intake have not yet been evaluated in a clinical trial, hydralazine has been shown to possess salubrious actions in addition to its anti-hypertensive activity. Administration of hydralazine has been shown to be effective in suppressing plasma cholesterol level[58]. Sharply reduced the elevated levels of blood urea nitrogen, creatinine, cholesterol, and triglycerides were noted in streptozotocin-induced diabetic mice[59]. In streptozotocin-induced diabetic rats, hydralazine significantly reduced serum levels of triglycerides, phospholipids, free fatty acids, and cholesterol by an obscure mechanism[60]. Administration of hydralazine for three weeks has been reported to be beneficial in protecting dopaminergic neurons and recovering the locomotor performance in an MPTP-induced Parkinson's mouse model as well[19]. Altogether, these reports support our findings and indicate conserved health benefits for hydralazine therapy across evolutionary boundaries. Although more work needs to be done to elucidate hydralazine's mechanism(s) of action and its metabolic benefits in higher organisms, we suggest that activation of PKA signaling is a unifying mechanism explaining many of the beneficial effects observed for this drug.

Hydralazine possesses an unusual set of properties that make it a viable candidate as an anti-aging molecule as well as therapeutic agent for the treatment of age-related diseases. The unique properties of hydralazine include but are not limited to: (1) carbonyl chelation which is protective against reactive carbonyl species, for example, to inhibit acrolein-mediated damage during spinal cord injury, (2) activation of PKA, sirtuins and NRF2 known to have profound impact on mitochondrial function, antioxidant homeostasis, and cellular health and stress protection which deteriorates with age, (3) antihypertensive and cardiovascular benefits, critical health factors in elderly populations. Choosing suitable indication(s) and correct dosage is important for attaining therapeutic effects and avoiding potential side effects considering hydralazine's complex mode of action and target multiplicity. Our results also call for an assessment of the potential anti-aging function of hydralazine in higher organisms. Hydralazine may deserve a fair trial to assess its place among other known FDA approved drugs with anti-aging benefits such as metformin, rapamycin, lithium, and the plant-derived supplement, resveratrol.

## Methods

**Chemicals**. All the chemicals were purchased from Sigma (St. Louis, MO) unless otherwise stated.

**Cell culture**. Neuroblastoma SH-SY5Y (ATCC CRL-2266) and myoblast C2C12 (ATCC® CRL-1772™) cells were purchased from ATCC. Cells were maintained in DMEM medium supplemented with 10% fetal bovine serum and were cultured in a humidified chamber at 37 °C with 5% $CO_2$. Cells were plated the day before treatment so that the density of cells could reach ~80% confluency at the time of harvest for final assays. Hydralazine was diluted in culture medium from an aqueous stock solution. The final concentration of hydralazine and the duration of the treatment were indicated in the text and the figure legends.

**Establishment of stable knock-down cell lines**. SIRT1 and SIRT5 were knocked down with specific shRNA in TRC Lentiviral Plasmid Vectors (Sigma). Human-specific sequence of Sirt1 (NM_012238.3-872s21c1, CCGGCAGGTCAAGGGAT GGTATTTACTCGAGTAAATACCATCCCTTGACCTGTTTTTG) and Sirt5 (NM_012241.2-762s21c1 CCGGGAGATCCATGGTAGCTTATTTCTCGAGA AATAAGCTACCATGGATCTCTTTTTG) were used for SH-SY5Y cells. A scrambled shRNA was used as a negative control. For C2C12 cells, mouse-specific shRNAs were used to target sirt1 (NM_019812.2-1370s21c1, CCGGAGTGAGAC CAGTAGCACTAATCTCGAGATTAGTGCTACTGGTCTCACTTTTTTG) and sirt5 (NM_178848.2-84s1c1, CCGGCCAGTTGTGTTGTAGACGAAACTCGAG TTTCGTCTACAACACAACTGGTTTTTG). Plasmids were extracted from the overnight grown bacteria using QIAprep Spin Miniprep Kit (27104, QIAGEN). Transfection was performed using FuGENE HD transfection reagent (Promega, E2311). DNA/reagent complex was prepared following the vendor instructions, then directly added to the 50% confluent cells. Transformed stable cells were selected using 0.8 μM puromycin for 2 weeks. Expression of SRIT1 and SIRT5 was determined by western blot analysis.

**Western blot analysis**. Protein abundance was determined by western blot analysis. At the end of the treatment, cells were collected and washed once in ice-cold PBS buffer, followed by lysis with RIPA buffer (1 mM EDTA, 1% Triton X-100, 0.5 % sodium deoxycholate, 0.1% SDS, 150 mM NaCl, 100 mM Tris-HCl, pH 7.2) supplemented with cocktails of proteases and phosphatases inhibitors (Thermo Fisher, Waltham, MA, USA) on ice. The cell lysates were then sonicated and centrifuged at $12,000 \times g$ for 10 min at 4 °C. Supernatants were collected and protein concentrations were measured using the bicinchoninic acid (BCA) assay (Pierce, 23228). An equal amount of protein from each sample was run in Tris-glycine SDS-PAGE gel, followed by transfer to PVDF membrane. Membranes were blocked with 5% milk followed by overnight incubation with specific antibodies: NRF2 (NBP1-32822, 1/1000 dilution), from Novus Biologicals, GAPDH (sc-47724, 1/5000 dilution), PGC1A (H300, sc-13067, 1/1000 dilution), CPS1 (sc-376190, 1/ 1000 dilution) from Santa Cruz Biotechnology, β-Actin (MA5-15739-HRP, 1/2000 dilution), COXIV (A21348, 1/1000 dilution) from Thermo Fisher Scientific, Anti-GFP, N-terminal (G1544, 1/1000 dilution) from Sigma-Aldrich, ATPB (ab14730, 1/ 1000 dilution) from Abcam, SIRT1 (9475, 1/1000 dilution), SIRT5 (8782, 1/1000 dilution), Cytochrome c (CYCS) (4280, 1/1000 dilution), TFAM (7495, 1/1000 dilution) and acetylated-lysine (9441, 1/1000 dilution) from Cell Signaling Technology, cAMP protein kinase catalytic subunit (ab76238, 1/2000) from Abcam. The membrane was subsequently incubated with species-specific HRP-conjugated secondary antibody followed by incubation with chemiluminescence substrate and imaging. The band intensity of each of the target proteins was quantified using Image J software (GE Healthcare, Sweden). We presented the relative amount of proteins quantified by Western blot analysis as "protein abundance". The intensity of each protein was normalized to the corresponding internal control (actin or GAPDH), then the normalized values from treatments were divided by the corresponding control(s) to calculate the relative abundance of each protein.

Uncropped scans of the most important blots are provided as supplementary information.

**Immunoprecipitation (IP)**. Cells were lysed in ice-cold IP buffer (150 mm NaCl, 10 mM Tris-HCl (pH 7.4), 1 mm EDTA, 1 mM EGTA (pH 8), 15 Triton X-100, 0.5% NP-40, 10 mM NAM, 5 mM sodium butyrate, 1 μM Trichostatin A, and protease inhibitors), followed by centrifugation at $12,000 \times g$ for 15 min at 4 °C. The supernatant was used for IP with antibodies specific for PGC1A and CPS1, on a rotator with constant stirring at 4 °C overnight. Protein A/G magnetic beads (Thermo Fisher) were added into the antibody-antigen mixture followed by incubation for 1.5 h at 4 °C on a rotator. A magnetic stand was used to collect the beads, followed by washing in lysis buffer four times. The captured complex was eluted with 1X SDS electrophoresis sample buffer. The eluted protein was analyzed by Western blot analysis.

**Real-time quantitative RT-PCR**. The relative levels of target genes mRNAs were measured by qRT-PCR. DNA was isolated from SHY-SY5 cells using NucleoSpin Tissue (Clontech,740952). RNA was isolated from the cells using a Total RNA Mini Kit (Bio-Rad, 7326820) and reversely transcribed to cDNA using Maxima First Strand cDNA Synthesis Kit (Thermo Scientific, K1671). PowerUp SYBR Green Master mix (Thermo Fisher Scientific, A25742), specific primers (Sigma) of the target genes (Supplementary Table 2) and an equal amount of the diluted DNAs or cDNAs. Reactions were performed on a C1000 Thermal cycler (Bio-Rad) machine and data were analyzed by Bio-Rad CFX manager 3.1 software using the ΔΔCq method. Actin or GAPDH were used as internal controls to normalize the gene expression data.

**C. elegans strains and maintenance**. C. elegans strains were cultured at 20 °C on standard nematode growth media (NGM) agar plates seeded with E. coli strain HB101 following the standard conditions. N2 worms were used as wild-type and the following mutants and transgenic strains were also provided by the Caenorhabditis Genetics Center (University of Minnesota): CW152 gas-1(fc21) X, TK22 mev-1(kn1) III, MIR13 sir-2.1(ok434) IV; aak-2 (ok524) X, RB754 aak-2 (ok524) X, VC199 sir-2.1 (ok434) IV, EU31 skn-1(zu135) (IV)/ nT1[unc-?(n754); let-?] (IV;V), EU1 skn-1(zu67) (IV)/nT1[unc-?(n754);let-?] (IV;V), LG357 skn-1 (zu135) (IV)/nT1[qIs51] (IV;V);geIs10 [ges-1p(long)::skn-1c::GFP + rol-6(su1006)], BR5270 byIs161[Prab-3::F3DK280;Pmyo-2::mCherry], BR6516 byIs194;[Prab-3:: F3DK280(I277P)(I308P);Pmyo-2::mCherry], SJ4100 zcIs13 [Phsp-6::GFP], SJ4103 zcIs14 [Pmyo-3::GFP(mit)], CF512 rrf-3(b26) II; fem-1(hc17). No ethics was required for C. elegans experiments.

**Lifespan analysis**. All lifespan assays were performed at 20 °C unless otherwise is stated using HB101 as food source according to standard protocols. Embryos were obtained from the adult nematodes by hypochlorite solution. The synchronized L1 worms hatched overnight were transferred to NGM agar plates. Hydralazine was added freshly from a 5 mM stock to the NGM media. Water was used as the control. The L4 population of worms was randomly split to control or treatment groups in a density of about 20–30 worms per 6 cm plate dish (Corning Inc.). The first day of adulthood was considered day one. Worms were transferred to fresh plates every day after reaching adulthood, and every 2–3 days after reaching 8 days of age.

For high glucose experiments, glucose was added to the NGM plates from a stock solution (45% in $H_2O$) to reach the desired final concentrations in the plates, at least 1 h before transferring animals. All the glucose experiments performed in the dark to reduce the adverse effects of light on nematode lifespan[61]. 5′-fluorodeoxyuridine (FUDR, Cayman chemicals) at the final concentration of 40 μM was used for two days to sterilize the worms. Every worm was subjected to a prodding test with a worm pick to count the number of dead worms. The animals that crawled off the plate, ruptured, or died from internal hatching were censored. The survival curve was plotted using Prism 7 and the significance of the curves calculated by Log-rank (Mantel-Cox) test.

**RNA interference**. Synchronized rrf3/fem-1 worms were grown at 25 °C till adulthood. Young adult sterile animals were moved to NGM plates containing one mM IPTG and fed HT115 bacterial strain containing scrambled or kin-1 RNAi plasmids (Vidal library) at 20 °C. The clone was verified by sequencing.

**Rotenone stress test**. A synchronized population of C. elegans was treated for 3 days with hydralazine. Adult worms were transferred to fresh NGM plates either preloaded with rotenone or rotenone plus hydralazine. After 24 h, prodding test was performed to score the number of dead animals.

**Locomotion assays**. Locomotor performance of the worms was assessed by measuring the number of body-bend. Worms were subjected to 30 s video recording on a Zeiss Axio Zoom. V16 fluorescence dissecting microscope equipped with Axiocam 503 and ZEN2 software. The bending rate was measured by placing live animals on a plate containing M9 buffer, allowing them to recover for 2 min, followed by filming for 30 s and counting the number of bends.

**Fluorescence microscopic imaging**. Synchronized populations of animals (zcIs14) were mounted on slides with 0.5 mM levamisole (Sigma) to induce muscle paralysis to study the morphology of the mitochondria. Worms were visualized with a Zeiss AxioImager M2 microscope equipped with a Hamamatsu Flash 4.0 Scientific c-mos camera and Zen2 software (×40). Images were taken with the same exposure and were processed and analyzed identically. The hsp6::GFP intensity was measured similarly with ×5 magnification and the quantification was done by ImageJ using the whole worm signal.

**Cell viability assay and ATP measurement**. Cells were cultured in 96-well white wall clear bottom plates. Cell viability assay was performed using the CellTiter-Glo reagent (Promega, G7571). At the end of treatment, 12 μl of the ½ times diluted reagents added into each well in 96 well plates. The luminescence signal was recorded using a Spectramax Gemini XPS plate reader (Molecular Devices, Sunnyvale, CA), after 30 min of incubation at room temperature. ATP measurement was done similarly as described above and data were normalized to the number of the cells.

**Mitochondrial membrane potential assays**. Changes in mitochondrial membrane potential in live cells quantified by flow cytometry (FCM) and microplate spectrophotometry. Cells were grown in 12-well plates. Tetramethylrhodamine, ethyl ester (TMRE, T669, Thermo Fisher Scientific) dissolved in DMSO and diluted to the final concentration of 10 nM in media. Old media was replaced with fresh media containing TMRE. After 45 min incubation, cells were removed with trypsin, washed once and suspended in PBS. FCM analysis was performed using FACS-Calibur instrument at Flow cytometry core facility of UTSWMC. Data analysis was done by quantification of the signal of at least 10,000 cells using Flowjo software package. For microplate spectrophotometry quantification of mitochondrial membrane potential, cells grown in 96-well black wall clear bottom plates. Cell staining was performed as described above, the media was removed, and cells were washed with PBS. TMRE signal was measured at Ex/Em = 549/575 nm using Spectramax Gemini XPS plate reader (Molecular Devices, Sunnyvale, CA). FCCP (carbonyl cyanide 4-(trifluoromethoxy) phenylhydrazone) treatment (10 μM, 30 min) was used as a control to eliminate mitochondrial membrane potential and TMRE staining. Data were normalized to the protein content of the cells in wells measured using BCA protein assay.

To measure mitochondrial membrane potential in live C. elegans, worms were recovered from the plates in M9 buffer, washed three times to remove residual bacteria, and resuspended in 250 μl of the diluted TMRE (100 μm) solution in M9 for 45 min. Worms were then washed five times with 1 ml of M9 buffer to eliminate the TMRE reagent and then transferred into a black-walled 96-well plate for reading. The fluorescence signal produced by the TMRE from ~50 worms was measured using a microplate reader, as described above.

**Mitochondrial complexes activity**. Cells were grown and treated in 15 cm dishes (Corning Inc.). After treatment, cells were removed with trypsin and washed once using ice-cold PBS buffer. Isolation of mitochondria was performed by Mitochondria Isolation Kit for Cultured Cells (Abcam, ab110170) following the manufacturer instructions. In-gel activity assay was performed to evaluate mitochondrial complex I and IV activity. The isolated mitochondria were solubilized in cold 1X NativePAGE sample buffer (Thermo Fisher Scientific) containing 1% digitonin by pipetting up and down and by inversion. After 15 min incubation on ice, lysates were centrifuged at $20,000 \times g$ for 30 min at 4 °C. The supernatant protein concentration was determined using BCA protein assay. After adding the NativePAGE 5% G-250 sample additive, an equal amount of the samples were loaded on the 3–12% gradient NativePAGE Novex Bis-Tris gels (Thermo Fisher Scientific) and the electrophoresis was performed following the vendor instructions. The upper part of the gels was incubated with complex I buffer consisting of 5 mM Tris/HCl (pH 7.4), 0.1 mg/mL NADH, and 2.5 mg/mL Nitro Blue tetrazolium chloride (NBT). The lower slice of the gels was incubated with complex IV buffer, containing 5 mg 3,3′-diaminobenzidine tetrahydrochloride (DAB) in 9 ml 50 mM phosphate buffer pH 7.4, 20 μg/ml catalase, 10 mg cytochrome c, and 750 mg sucrose for complex IV staining[62]. All the catalytic staining reactions were performed at room temperature.

The activity of mitochondrial complex I and IV was also assessed spectrophotometrically. The activity of the mitochondrial complexes I was measured using the isolated mitochondria from the control and treated cells with Complex I Activity Assay Kit (Cayman Chemical, 700930). After adding the isolated mitochondria to the prepared reagents, as described by the vendor, immediately the absorbance was measured at 340 nm (30 s intervals for 20 min at room temperature) using a 96-well plate reader. Time-dependent reaction absorbance was plotted versus time, and the reaction rate was calculated based on the slope of the linear portion of the curve. Similarly, the activity of complex IV was determined using Complex IV Activity Assay Kit (Cayman Chemical, 700990). The isolated mitochondrial samples were added to the buffer containing complex IV activity assay and reduced cytochrome c assay reagent in 96-well plates, and the absorbance was measured immediately at 550 nm (30 s intervals for 20 min at room temperature). Mitochondrial complex II activity was determined using the Complex II Enzyme Activity Micropale Assay Kit (Abcam, ab109908). In-well

purification of the mitochondrial complex II was performed from the control and treated cell lysates using premium NuncMaxiSorp™ modular microplates, following the manufacturer standard protocol. After the complex II was immobilized in the well, the substrate was added and enzyme activity was measured as the change in absorbance (600 nm) at 1-min intervals for 1 h at room temperature with no plate shaking. The activity of the complex was calculated based on the ratio between two time points for all the samples where the decrease in absorbance was the most linear. MitoTox Complex V OXPHOS Activity Microplate Assay (Abcam, ab109907) was used to measure the activity of the ATP synthase complex, as explained by the manufacturer. Detergent-solubilized mitochondria isolated from control and treated cells, as explained above, were added to the provided microplates to immunocapture complex V. Phospholipids added to the plate, and the absorbance was measured at 340 nm in kinetic mode for one hour at 30 °C. Complex V activity calculated proportionally to the decrease in absorbance at 340 nm.

**Seahorse respiration assays.** OCR and extracellular acidification rate (ECAR) were measured using a Seahorse XF-24 Flux Analyzer (Seahorse Biosciences, North Billerica, MA, USA). Cells were seeded at an appropriate density in Seahorse cell culture plates to reach approximately 80% confluency at the time of the experiment. A sensor cartridge was hydrated in Seahorse XF Calibrant at 37 °C in a non-$CO_2$ incubator overnight. The cells were washed in buffered DMEM, and the medium was then changed to the actual assay medium, prepared by supplementing Seahorse XF Base Medium with 1 mM sodium pyruvate, 2 mM glutamine, and 5 mM glucose, pH 7.4. Cells were incubated in 37 °C incubator without $CO_2$ for 45 min to one hour before the assay to allow pre-equilibration with the assay medium. OCR was detected under basal conditions followed by sequential addition of oligomycin, FCCP, as well as rotenone & antimycin A to the final concentrations of 1, 1, 1.5, and 1 μM respectively. OCR values from each well were normalized to the amount of protein measured by BCA protein assay.

Respiration of the nematodes was measured using the Seahorse XF24 equipment (Seahorse Bioscience Inc.) as described elsewhere[63]. Animals were recovered from the plates and washed three times in 1 mL M9 to eliminate residual bacteria, and resuspended in 0.5 ml M9 medium. Worms were transferred in 24-well standard Seahorse plates with about 50 worms per well, and oxygen consumption was measured six times. Respiration rates were normalized to the number of worms in each well.

**SIRT1 enzyme activity assay.** C2C12 cells grown in 10 cm dishes were used to measure SIRT1 activity. Cells were lysed in a buffer (10 mM HEPES KOH (pH 7.5), 420 mM NaCl, 0.5 mM EDTA, 0.1 mM EGTA, 10% glycerol) with sonication for 30 s. After 30 min incubation on ice, samples were centrifuged at $20,000 \times g$ for 10 min and the supernatant was used for the reaction. Protein content was measured by BCA assay, and an equal amount of the cell lysates were used for enzyme activity measurement using a commercial SIRT1 Activity Assay Kit (Abcam, ab156065). SIRT1 activity quantified based on the fluorescent signal emitted from the fluoro-substrate peptide using a Spectramax Gemini XPS plate reader (Molecular Devices) at Ex/Em = 350/460 nm, following the vendor instructions.

**PKA kinase assay with radiolabeled ATP.** Control and PKAcat knock out HEK293 cells were treated with hydralazine for 2 h. Cells were lysed in lysis buffer (50 mM Hepes, pH 7.7, 1.5 mM $MgCl_2$, 0.2 mM $Na_3VO_4$, 1 mM EGTA, 150 mM NaCl, 10% Glycerol, 100 mM NaF, 50 mM beta Glycerophosphate, 0.1% NP40) containing a protease inhibitor cocktail via sonication, 2× for 10 s. Cell debris was removed by centrifugation ($15,000 \times g$, 10 min) at 4 °C and supernatants were used for PKA kinase assays after protein measurement. Reactions were initiated by adding 20 microgram histone 2B (protein substrate), 50 micromolar ATP ([γ −32P] ATP 10,000 cpm/pmol), 10 mM $MgCl_2$, 10 mM HEPES pH 8.0, 1 mM DTT and 1 mM benzamidine for 1 h at room temperature[64]. The reaction was stopped by adding 5x Laemmli sample buffer followed by heating at 100 °C. Samples were loaded on 4–12% Crit XT Bis-Tris gels (Bio-Rad). Gel staining, drying and scintillation counting were performed[64]. The histone band from the lane lacking cell extract was subtracted to correct for background.

**Metabolomics analysis.** A population of about 100 wild-type *C. elegans* per condition was recovered from NGM plates with M9 medium and washed three times in 1 mL M9. After removing the extra M9 buffer, samples placed on dry ice and 0.5 ml of 80% (v/v) methanol (cooled to −80 °C), containing heavy standard amino acids were added, and the tubes were incubated at −80 °C for 20 min. Worms were lysed by sonication at 4 °C for five times with interval incubation at −80 °C for 5 min. Samples were centrifuged at $14,000 \times g$ for 15 min at 4 °C to pellet the cell debris. The metabolite-containing supernatant filtered and transferred to new tubes for MS analysis. The pellet used for protein measurement using BCA assay. Mass spectrometry analysis was performed according to previously published work[65]. 6500 QTRAP mass spectrometer (AB SCIEX) equipped with an ESI ion spray source operated in both positive and negative ion modes were used for mass spectral analysis. The mass spectrometer was coupled to an HPLC system (Shimadzu). The system was controlled by Analyst 1.5 software (AB SCIEX, Framingham, MA). Selected reaction monitoring (SRM) mode was used to detect the

multiple metabolites as they eluted off of the chromatographic column using a dwell time of 100 ms per metabolite. The ion spray needle voltages set at 3600 V and −3600 V, for positive and negative modes, respectively. ZIC-pHILIC (5 μm, polymer) PEEK 150 × 2.1 mm column (Millipore-Sigma) with the flow rate of 150 μL/min and column temperature of 4 °C. The chromatography conditions were as follows: Solvent A: 20 mM ammonium carbonate and 0.1% $NH_4OH$ in water; Solvent B: acetonitrile. The elution gradient was 0 min 80% B, 20 min 20% B, 20.5 min 80% B, 34 min 80% B for both positive and negative mode of SRM. The MultiQuant 3.0.3 software (AB SCIEX) was used to integrate the SRM peak areas across the chromatographic elution. Data were imported to MarkerView 1.3.1 (AB SCIEX) for further analysis. Data normalization was performed based on the protein content of each sample, and the Student's $t$-test was used to evaluate the statistical significance of metabolite intensity differences between control and treatments. MetaboAnalyst 4.0 used for further analysis of the data[66]. The significantly changed metabolites were subjected to metabolite set enrichment analysis (MSEA) and pathway analysis using MetaboAnalyst 4.0. Over representation analysis (ORA) was implemented using the hypergeometric test to evaluate if a particular metabolite set is represented more than expected by chance within the compound list. The heat map of the metabolites was plotted using MetaboAnalyst 4.0.

**Thermal proteome profiling in *C. elegans*.** Wild-type *C. elegans* were used for TPP experiment. Animals were treated with hydralazine (100 μM) for 2 h then collected and washed three times using M9 buffer. Control and treated groups were equally dispensed to 10 PCR tubes using PBS with about 80 worms in each tube. Heat treatment was done using a gradient from 37 to 67 °C for 3.5 min in a thermocycler machine[38]. Worms were transferred to 1.5 ml tubes after adding PBS containing 0.08 NP-40 and protease inhibitor cocktail (Thermo Scientific) to a final volume of 70 μl. Protein extraction was done with three freeze-thaw cycles using liquid nitrogen followed by sonication. Soluble proteins were separated by centrifugation for 30 min at $100,000 \times g$, and supernatant were used for further analysis. Protein concentration was measured by BCA assay and proteins digested using S-Trap (Protifi) workflow following the vendor instruction. Peptides were labeled using TMT- 10 plex (Thermo-Fisher Scientific) as described by the vendor. After quenching the reactions, TMT-labeled samples were mixed in equal ratio followed by clean up with C18 solid-phase extraction (SPE) cartridge (Sep-Pak, Waters). Peptides were reconstituted in 2% acetonitrile and 0.1% TFA. MS analysis was performed using a Fusion Orbitrap Lumos mass spectrometer (Thermo Scientific) connected to a Dionex Ultimate 3000 UHPLC (Thermo Scientific). An Easy-Spray column (75 μm × 50 cm) packed with 2 μm C18 material was used for chromatographic separation. Gradient elution was performed from 2% acetonitrile to 40% acetonitrile in 0.1% formic acid over 120 min. The mass spectrometer was set to acquire data in data-dependent top 10 method using a resolution of 120,000. Proteomics searches were done using *C. elegans* reference database from UniProt. Peptides with a minimum of six amino acid residues and a maximum of two miscleavages with trypsin/P specificity were identified. Methionine oxidation was used as a variable modification and protein N-terminal acetylation and carbamidomethyl of cysteine were used as fixed modifications. Search tolerance was set at 10 ppm for MS1 precursor ions and 0.5 Da for MS2 fragment ions. Protein and peptide false discovery rate were set at 1%. At least one unique or razor peptide was required for protein identification. Unique plus razor peptides were used for quantification. Data processing was performed following the data analysis workflow described by Franken et al.[38].

**In silico simulation of hydralazine/PKA interaction.** The AutoDock v4 distribution was utilized for docking[67]. The chain A of the crystallographic 3D structure of cAMP-dependent protein kinase catalytic subunit alpha (PRKACA) was used for flexible docking modeling. 2GU8 with a 2.2 Å of resolution and zero Ramachandran outliers was selected from PDB for the molecular modeling. An empirical free-energy force field and rapid Lamarckian genetic algorithm search method used to predict the bound conformations. Docking strategy was performed in order to predict the preferred orientation of one molecule (e.g., Hydralazine as a ligand) to a second (e.g., PRKACA as the target). A flexible docking strategy was used where both molecules were allowed to change their conformation. Although hydralazine has a rigid structure, we did not want to force it in a specific position. To select the best pose, we focused on the prediction of various poses (i.e., candidate binding mode) in a blind mode when ligand and receptor bind to each other. A scoring function used to rank all poses based on calculated binding energies[68].

**Statistical analysis.** Raw data acquired using gel imaging (Western blot and mitochondrial complex activity measurements), flow cytometry (mitochondrial membrane potential measurements), Seahorse (oxygen consumption assays), spectrophotometry (mitochondrial complex activity, ATP assay, viability assay, mitochondrial membrane potential, and SIRT1 enzyme activity measurements), qPCR (gene expression, mtDNA/nDNA ratio measurements), microscopy (locomotor performance measurements), manual counter (animal survival measurements), and mass spectrometry (relative metabolite quantification) were processed using GraphPad Prism 7 software. Two-tailed Student $t$-test was used to assign significance with one star(*) representing $p \leq 0.05$ and two stars (**) representing

$p \leq 0.01$. N refers to the number of animals (*C. elegans*), while $n$ stands for the number of independent biological replicates or the number of individual populations of animals. The significance of lifespan curves was calculated by Log-rank (Mantel-Cox) test using GraphPad Prism 7 software.

## Data availability

Data supporting the findings of this study are provided either in the Supplementary Information file or in the Source Data file (data for Figs. 1a, 1b, 1d, 1e, 1f, 1g, 1h, 1i, 2a, 2c, 2f, 2g, 3i, 4b, 4c, 4d, 4f, 4g, 4h, 5b, 5d, 5f, 5g, 5h, 6a–c, supplementary Fig. 1a, f and supplementary Fig. 4b, d are provided in the Source Data file). Proteomics data have been deposited in Proteomexchange (accession code PXD005618) and MassIVE (accession code MSV000084074). All data are available from the authors upon reasonable request.

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

## Acknowledgements

We thank Drs. Melanie Cobb, David Mangelsdorf, and Nima Sharifi for advice on our manuscript. We thank Dr. John Hulleman for providing us with constructs and reagents. Special thanks to Steve Stippec for his assistance with PKA kinase assays. We thank Dr. Jenna Jewell for providing us with a PKA KO cell line. We also thank Mrs. Sophie Guo with her assistance with Western blot analysis, Dr. Amirmanssor Hakimi for help with DAVID analyses, and Dr. Hamid Baniasadi for metabolomics analyses. We thank Dr. Asish Chaudhuri for general advice. All *C. elegans* strains were provided by CGC, which is funded by the NIH Office of Research and Infrastructure Programs (P40OD010440). This work was supported by the Robert A. Welch Foundation (grant I-1850 to H.M.) and the Cancer Prevention and Research Institute of Texas (grant R1121 to H.M.). H.M. is the founder of Neurodaroo LLC.

## Author contributions

E.D. designed and conducted the experiments and wrote the manuscript. M.G. performed in silico docking simulation and data analysis of TPP experiments, B.S. helped with flow cytometry and Seahorse analysis. R.L. provided advice and support for all *C. elegans* studies. H.M. designed and supervised the project and wrote the manuscript.

## Competing interests

The authors declare no competing interests.
