## [Peer Review File · Nature Communications]

Reviewers' Comments:

Reviewer #1:

Remarks to the Author:

The manuscript reports that hydralazine, an FDA-approved drug that has been in use for many decades, for a broad range of indications, improves mitochondrial function via sirtuin activation, and this is demonstrated using cell culture and the model nematode *C. elegans*. In a previous paper (published 2017) the same authors had reported that hydralazine induces stress resistance and extends *C. elegans* lifespan by activating the NRF2/SKN-1 signaling pathway. In the 2017 paper, it was shown that hydralazine restores viability of primary neuronal cells stressed with rotenone, a mitochondrial complex I inhibitor, and that hydralazine also protects *C. elegans* against rotenone toxicity. In the current manuscript, the effects of hydralazine on the mitochondria is studied in depth and it is shown that improvement of mitochondrial function by hydralazine requires SIRT1/sir-2.1 and SIRT5.

The manuscript starts off with an exploration of the effects of hydralazine on mitochondria, summarized in Figure 1. Then, data shown in Figure 2 convincingly demonstrate that SIRT1 and possibly SIRT5 are required for the effects of hydralazine on mitochondrial function. In Figure 3, panel a shows that hydralazine still has an effect on a *C. elegans* model of tauopathy; however, I don't agree that the data show that hydralazine "restores" normal mitochondrial function. Both arms (anti- and pro-agg) show similar effect size for treatment with hydralazine, but it does not appear that hydralazine makes Pro-agg behave like Anti-agg, or reach a mito membrane potential similar to that of untreated anti-agg. These two conditions also aren't compared for significance (as an aside, significance levels, in this panel and many others, are rather modest, which isn't a necessarily problem when there are many corroborating lines of evidence). In panel 3b, it is shown that SIRT1 KD or SIRT5 KD prevents rescue of AP cells by hydralazine, indicating that SIRT1 is required for hydralazine-mediated improvement of viability which is then further supported by additional experiments.

Similarly, it is claimed that data in Figure 5a show that hydralazine restores normal lifespan to *C. elegans* grown under high glucose conditions. However, the lifespan curves show that the effects of hydralazine on worm lifespan under high glucose are very similar to the effects observed for normal conditions, thus these data do not provide evidence that hydralazine directly interacts with the pathways underlying glucose toxicity. Rather, it appears that hydralazine affects lifespan, mobility, and metabolism also at glucose levels, similar to other stress conditions. Therefore, the data on high glucose levels, in their entirety, seem distracting from the mechanistic aspects of the paper and should be removed (or perhaps moved to the SI).

In conclusion, that sirtuins are required for the effects of hydralazine on mitochondria is interesting. Sirtuins here presumably act upstream of NRF2, though this could be made clearer in the manuscript, and alternative pathways connecting sirtuins and NRF2, e.g. NRF2 activation via ROS-generated from oxidative catabolism of sirtuin-generated nicotinamide (see Schmeisser Nat Chem Biol 2013) should be included in the discussion.

Nonetheless, I'm not enthusiastic about this study, for two reasons.

- beneficial effects of hydralazine on healthspan, stress resistance, and mitochondrial function were already reported in the 2017 paper. Much of the current study feels more of an extension of this previous work
- though involvement of sirtuins is clearly demonstrated, their (biochemical) roles or specific functions are not clear. Moreover, one obvious and most interesting question is not addressed – what proteins/metabolites does hydralazine interact with to improve mito function and extend healthspan? Unless this latter point can be addressed I do not believe this study merits publication in Nature Communications.

Minor points:

- "exponential" on page 3 is probably not correct, in the scientific sense of the word.
- there are a few spelling and punctuation mistakes in the legend to Figure 3

- page 11: the sentence "We only observed rotenone resistance in 3 SIR-2.1 mutant worms" is misleading – what's observed is that hydralazine does not confer rotenone resistance in sir-2.1 worms.

Reviewer #2:

Remarks to the Author:

This study is a direct extension of a previous publication by this group (Nature Communications 2017) in which improved mitochondrial function and prolonged lifespan of *C. elegans* due to NRF2 signaling upon hydralazine treatment was reported. In the present manuscript Dehghan and co-workers now report a supplementary mechanism of hydralazine-induced SIRT1/ SIRT5 activation which contributes to this phenotype. More specifically they now demonstrate that hydralazine improves carbon metabolism, increases TCA cycle metabolites and improves respiration in SIRT1/ SIRT5 dependent manner.

Questions

Major phenotypes – improved mitochondrial function and prolongs life-span of *C. elegans* upon hydralazine treatment were already reported in the previous study, albeit with a different mechanism. How does hydralazine-induced NRF2 signaling relate to SIRT1/SIRT5 signaling? How does hydralazine induce such distinct responses? Is there a hierarchical order of hydralazine responses which would explain induction of these specific responses, i.e. could promoter methylation be a unifying mechanism?

Hydralazine has been in clinical use for more than 50 years, and also for hypertension and heart failure in patients with diabetes mellitus. Is there evidence for a beneficial role of hydralazine on metabolism in diabetic patients? To my knowledge such association has been observed for metformin, but not for hydralazine?

Minor questions:

Figure 1: How do concentrations used of 5-20μM relate to doses used in patients and vertebrate studies?

Figure 2: Data presented in panel b is unimpressive and leaves some doubt if this is truly biologically relevant. The authors should employ an alternate method of detection to corroborate these findings. Panel h suggests an association of NRF2 and SIRT1 up-regulation. Does NRF2 overexpression induce SIRT1?

Figure 4 convincingly demonstrates an impact of hydralazine on multiple metabolic pathways. Does hydralazine also impact systemic glucose levels? Is there evidence for an impact of hydralazine on blood glucose levels and metabolic indicators in diabetic patients?

Supplementary Figure 1. Lack of an impact of HIF1 on hydralazine-induced SIRT1 signaling is somewhat unexpected. What is the impact of hydralazine in the *C. elegans* system? Is this cell culture system sufficient to study HIF1α?

Reviewer #3:

Remarks to the Author:

NCOMMS-18-33749-T "Hydralazine Improves Mitochondrial Function via SIRT1/SIRT5 Activation and Protects against Glucose Toxicity"

Strengths of the manuscript

The study is interesting.

Weaknesses of the work

The text is difficult to follow. Several methodological details are missing and/or are presented in a very confuse manner.

General comments

Dr. Dehghan and colleagues showed that hydralazine (a FDA approved drug) has anti-aging proprieties via the activation of the NRF2/SKN-1 signaling pathway in a previous study (Dehghan E. et al. Hydralazine induces stress resistance and extends *C. elegans* lifespan by activating the NRF2/SKN-1 signalling pathway. *Nat Commun* 8(1):2223, 2017). The present work continues and deepens the molecular and metabolic mechanisms of hydralazine. The authors show that hydralazine improves mitochondrial function and metabolism in a SIRT1/SIRT5-dependent manner. Moreover, they described that this molecule is able to protect *C. elegans* against high glucose toxicity. The present manuscript underlines that the mitochondrial effects of hydralazine are essentials for its anti-aging properties.

Although this manuscript describes several interesting findings, some major and minor points should be addressed (listed below in no particular order).

- The cell death and cell viability evaluation should be corroborated:

(a) Figures 1a, 2e, 2f, 3a, 6b: the authors should evaluate cell death (PI or DAPI staining) in these experimental conditions.

(b) Figure 3b: the cell viability has been evaluated with the CellTiter-Glo assay (Promega, G7571) (pg. 40, lines 4-11). This method determines the number of viable cells in culture based on quantitation of the ATP present (as an indicator of metabolically active cells). Because the authors demonstrated that Hydralazine modulates SIRT1 and SIRT5 and activates mitochondria, they should evaluate the effect on viability with an assay independent to mitochondria.

(c) Discussion paragraph: The authors should take into account the paper of Ruiz-Magana MJ et al. "The antihypertensive drug hydralazine activates the intrinsic pathway of apoptosis and causes DNA damage in leukemic T cells. *Oncotarget*. 2016;7(16):21875-86. doi: 10.18632/oncotarget.7871.

- Figures 1b, 1d-1g, 1i, 2g: internal positive and negative controls are missing in these experimental settings.

- In my opinion, in order to simplify the reading of the manuscript, the figures should be made in a homogeneous manner. For example in figure 1a, the data are shown as "Mitochondrial membrane potential (% Ctrl)", whereas the same type of experiment is shown as "mitochondrial membrane potential (RFU)" in figures 2e and 2f. Often, the ordinate axis is different for similar comparable experimental settings (i.e. figures 1d-1g or figures 4d-e/figure 5e).

- Legend of figure 1h: the description of bars graph of figure 1h is missing (but probably is linked with legend of figure 1i?).

- Results paragraph (lines 12-14) concerning figure 2h: a mitochondrial marker, cytochrome c oxidase subunit IV (COXIV) is described in the text but it is missing in the figure.

- Figures 2b/2h: In Figure 2h, in "Ctrl(shRNA)" condition, 10uM of hydralazine treatment increases SIRT1 levels in a very significant manner, however in figure 2b the effect of the same concentration of hydralazine in normal cells is not so evident (at least in the representative SIRT1 w. blot shown). The time of Hydralazine (10uM) treatment is the same in figures 2b and 2h? (This information is missing in the figure legend).

- In general, it is not clear, for all the w. blot experiments, how the normalization (vs. the loading control) has been done. Which is the meaning of "Relative protein abundance (% Ctrl)" (the information is missing in the Materials & Methods paragraph)? Particularly, in figure 5f the actin bands shown are not convincing, whereas GFP levels are the same in all the experimental conditions.

- Figure 3d: The AN cells (as AP cells) have been treated with Hydralazine? Moreover, the COXIV w. blot shown is not so representative of the sentence "key regulatory elements of mitochondrial function..... were adversely affected by tau aggregates" (the levels of COXIV in AN and AP in control conditions seem the same).

- Figure 3f: In the graph, a protection was observed against rotenone in AAK-2 mutants treated with hydralazine and not as written in the text (p. 11 lines 2-3) "We only observed rotenone resistance in SIR-2.1 mutant worms (Fig. 3f)". The authors should clarify this point.
- Figure 4i: the authors should describe in the text the (significant) results obtained with the skn-1(zu67) mutant *C. elegans* as well as the meaning of the utilization of several skn-1 mutants.
- Legend of figure 5f: the size bar meaning is missing.
- Typo "improeed", p.16 line 13
- Legend of figure 6e: the legend of this part of the figure should be improved.
- A statistic paragraph should be added to the manuscript.

Reviewer #1 (Remarks to the Author):

The manuscript reports that hydralazine, an FDA-approved drug that has been in use for many decades, for a broad range of indications, improves mitochondrial function via sirtuin activation, and this is demonstrated using cell culture and the model nematode *C. elegans*. In a previous paper (published 2017) the same authors had reported that hydralazine induces stress resistance and extends *C. elegans* lifespan by activating the NRF2/SKN-1 signaling pathway. In the 2017 paper, it was shown that hydralazine restores viability of primary neuronal cells stressed with rotenone, a mitochondrial complex I inhibitor, and that hydralazine also protects *C. elegans* against rotenone toxicity. In the current manuscript, the effects of hydralazine on the mitochondria is studied in depth and it is shown that improvement of mitochondrial function by hydralazine requires SIRT1/sir-2.1 and SIRT5. The manuscript starts off with an exploration of the effects of hydralazine on mitochondria, summarized in Figure 1. Then, data shown in Figure 2 convincingly demonstrate that SIRT1 and possibly SIRT5 are required for the effects of hydralazine on mitochondrial function.

1. In Figure 3, panel a shows that hydralazine still has an effect on a *C. elegans* model of tauopathy; however, I don't agree that the data show that hydralazine "restores" normal mitochondrial function. Both arms (anti- and pro-agg) show similar effect size for treatment with hydralazine, but it does not appear that hydralazine makes Pro-agg behave like Anti-agg, or reach a mito membrane potential similar to that of untreated anti-agg. These two conditions also aren't compared for significance (as an aside, significance levels, in this panel and many others, are rather modest, which isn't necessarily a problem when there are many corroborating lines of evidence).

We concur with the reviewer's comment and have now replaced our previous statement which was "A significant reduction in $\Delta\Psi_m$ was observed in the pro-aggregant strain compared to the control (Fig. 4f). Hydralazine treatment, for four days, restored the $\Delta\Psi_m$ to a level comparable to the control" with the following "A significant reduction in $\Delta\Psi_m$ was observed in the pro-aggregant strain compared to the anti-aggregant strain (Fig. 4f). $\Delta\Psi_m$ measured in pro-aggregant treated cells was significantly higher than untreated cells." (now Page 13, lines 11 to 14)

2. In panel 3b, it is shown that SIRT1 KD or SIRT5 KD prevents rescue of AP cells by hydralazine, indicating that SIRT1 is required for hydralazine-mediated improvement of viability which is then further supported by additional experiments. Similarly, it is claimed that data in Figure 5a show that hydralazine restores normal lifespan to *C. elegans* grown under high glucose conditions. However, the lifespan curves show that the effects of hydralazine on worm lifespan under high glucose are very similar to the effects observed for normal conditions, thus these data do not provide evidence that hydralazine directly interacts with the pathways underlying glucose toxicity. Rather, it appears that hydralazine affects lifespan, mobility, and metabolism also at glucose levels, similar to other stress conditions. Therefore, the data on high glucose levels, in their entirety, seem distracting from the mechanistic aspects of the paper and should be removed (or perhaps moved to the SI).

Our aim in using a high-glucose diet was to induce mitochondrial dysfunction which underlies the etiology of many age-related diseases to show that hydralazine can protect mitochondrial function under various stressors. To address the reviewer's concern and because glucose was not intended to be a major focus, one part of the data in the glucose figure was moved to the supplement and the rest were combined with the data from other stressors in figure 4. We think this organization separates data on stressors from mechanistic data and reduces unintended distraction.

3. In conclusion, that sirtuins are required for the effects of hydralazine on mitochondria is interesting. Sirtuins here presumably act upstream of NRF2, though this could be made clearer in the manuscript, and alternative pathways connecting sirtuins and NRF2, e.g. NRF2 activation via ROS-generated from oxidative catabolism of sirtuin-generated nicotinamide (see Schmeisser Nat. Chem. Biol. 2013) should be included in the discussion.

The role of NRF2/SKN-1 in regulation of ROS, generated via promotion of mitochondrial function is demonstrated in Figure 6e. Thank you for bringing the Schmeisser et al. (2013) report to our attention. Now it is cited in our discussion. We also added the following paragraph in the discussion to clarify SIRT1-NRF2 relationship better now that we have identified cyclic AMP-dependent protein kinase (PKA) as the direct target of hydralazine." We demonstrated that activation of PKA is necessary for upregulation of SIRT1 and NRF2 signaling pathways and consequently extension of lifespan and protection against rotenone toxicity (Fig. 4-5h). We now demonstrate a direct link between PKA and SIRT1. Hydralazine-induced NRF2 activation can also be explained via a PKA-dependent mechanism (Supplementary Fig. 4a). It has been shown that cAMP/PKA and SIRT1 are upstream regulators of the NRF2-ARE pathway in fasting mice and human hepatocytes. Although our preliminary data suggest a possible role of SIRT1 as an upstream activator of NRF2 (Supplementary Fig. 5c), we cannot rule out NRF2 activation via other PKA-dependent regulatory mechanisms such as CREB binding protein or GSK3B. Further studies in mammalian cells and *C. elegans* are required to unravel the exact mechanism by which PKA mediates activation of SIRT1 and NRF2." (Page 20-21)

Nonetheless, I'm not enthusiastic about this study, for two reasons.

1. Beneficial effects of hydralazine on healthspan, stress resistance, and mitochondrial function were already reported in the 2017 paper. Much of the current study feels more of an extension of this previous work. In the 2017 paper, we proposed, based on proteomics data, that mitochondrial activation was a possible additional mechanism underlying hydralazine-mediated health benefits; but in that paper there were no experimental tests of that hypothesis. Here, for the first time, we provide evidence, both *in vivo* and *in vitro*, demonstrating elevation of mitochondrial function with hydralazine treatment. Additionally, we provide data to show that hydralazine-mediated

mitochondrial activation is SIRT1/SIRT5 dependent; SIRT1 and SIRT5 are activated as a result of the direct binding of hydralazine to PKA.

Though involvement of sirtuins is clearly demonstrated, their (biochemical) roles or specific functions are not clear.

There is a large list of substrates which can be targeted by SIRT1 and SIRT5.

Therefore, their effects are broad; finding a specific substrate for these enzymes that may be responsible for the observed phenotypes, is beyond the scope of this manuscript.

2. Moreover, one obvious and most interesting question is not addressed – what proteins/metabolites does hydralazine interact with to improve mito function and extend healthspan? Unless this latter point can be addressed I do not believe this study merits publication in *Nature Communications*.

Using proteomics screening and a series of biological experiments, we found and verified the direct binding target of hydralazine as cAMP-dependent protein kinase (PKA) which explains the activation of SIRT1/NRF2, promotion of mitochondrial function, stress resistance, and extension of lifespan. We think this discovery along with our other data merits publication in *Nature Communications*.

Minor points:

- “exponential” on page 3 is probably not correct, in the scientific sense of the word

The reviewer’s point is valid. Exponential growth has a precise scientific definition which does not fit the growth rate of citizens over the age of 80 as we had stated previously. So, we replaced the word “exponential growth” with “shift in population average age”.

- there are a few spelling and punctuation mistakes in the legend to Figure 3

Quite a few panels have been shuffled around for two reasons; 1) the discovery of PKA as the target of hydralazine, and 2) recommendation of the reviewer’s regarding the glucose-related data. As a result, figure legends have changed significantly. Spelling and punctuation mistakes in the original figure 3 have been corrected.

- page 11: the sentence “We only observed rotenone resistance in 3 SIR-2.1 mutant worms” is misleading – what’s observed is that hydralazine does not confer rotenone resistance in sir-2.1 worms.

We agree with the reviewer that the abovementioned sentence was misleading. So, in the revised manuscript this sentence has been replaced with the following “We only observed rotenone resistance in wild-type and AAK-2 mutant worms indicating that hydralazine-mediated protection is AMPK-independent (Fig. 4h).”

Reviewer #2 (Remarks to the Author):

This study is a direct extension of a previous publication by this group (Nature Communications 2017) in which improved mitochondrial function and prolonged lifespan of *C. elegans* due to NRF2 signaling upon hydralazine treatment was reported. In the present manuscript Dehghan and co-workers now report a supplementary mechanism of hydralazine-induced SIRT1/ SIRT5 activation which contributes to this phenotype. More specifically they now demonstrate that hydralazine improves carbon metabolism, increases TCA cycle metabolites and improves respiration in SIRT1/ SIRT5 dependent manner. Major phenotypes – improved mitochondrial function and prolonged life-span of *C. elegans* upon hydralazine treatment were already reported in the previous study, albeit with a different mechanism.

Questions

1. How does hydralazine-induced NRF2 signaling relate to SIRT1/SIRT5 signaling? How does hydralazine induce such distinct responses? Is there a hierarchical order of hydralazine responses which would explain induction of these specific responses, i.e. could promoter methylation be a unifying mechanism?

Using proteomics screening and a series of biological experiments, we found and verified the direct binding target of hydralazine as cAMP-dependent protein kinase (PKA) which explains the activation of SIRT1/NRF2, promotion of mitochondrial function, stress resistance, and extension of lifespan, as noted in the response to Reiewer #1 (Q3). Our observations indicate that both NRF2 and sirtuins are downstream effectors of PKA, the direct binding target of hydralazine (Fig. 5f and Supplementary Fig. 4d). PKA has many targets and both NRF2 and sirtuins are among its known downstream targets. Cross-talk between PKA and NRF2 via SIRT1 has been reported (PMC3331675, PMC3880903). Our data show that SIRT1 is an activator of NRF2, in that it increases protein abundance (Supplementary Fig. 5c). Other SIRT1-independent regulatory elements have also been implicated in cross-talk between PKA and NRF2. For example, GSK3B, a known inhibitor of NRF2, can be inhibited directly by PKA-mediated phosphorylation (PMC17277). Another possible mechanism is activation of NRF2 via CREB signaling, an event we have demonstrated with hydralazine treatment (PMID: 11683914). However, more needs to be done to understand the details of SIRT1-dependent and independent regulation of NRF2 via hydralazine. We have summarized our conclusions in Figure 6e indicating that PKA activation is most likely the unifying mechanism.

2. Hydralazine has been in clinical use for more than 50 years, and also for hypertension and heart failure in patients with diabetes mellitus. Is there evidence for a beneficial role of hydralazine on metabolism in diabetic patients? To my knowledge such association has been observed for metformin, but not for hydralazine?

From a search of the literature, we did not find data tracking the effects of hydralazine on cell metabolism and bioenergetics in diabetic patients. We agree this would be a valuable question for future investigation.

Minor questions:

Figure 1: How do concentrations used of 5-20µM relate to doses used in patients and vertebrate studies?

There has been a wide range of hydralazine concentrations administered for different purposes in human patients and different model organisms. We mainly used 5 µM and 10 µM; these are low compared to the concentrations used in much of the published work on hydralazine. In general, 5-20 µM hydralazine generates a mild to medium vasorelaxation (PMCID:PMC1572994).

Figure 2: Data presented in panel b is unimpressive and leaves some doubt if this is truly biologically relevant. The authors should employ an alternate method of detection to corroborate these findings.

We now provide additional supporting data by measuring SIRT1 enzymatic activity directly and also by measuring the deacetylation of SIRT1 substrate PGC1A and SIRT5 substrate CPS1 (Figures 2c-e).

Figure 2: Data presented in panel h suggests an association of NRF2 and SIRT1 up-regulation. Does NRF2 overexpression induce SIRT1?

Our data and others support the upstream regulatory role of SIRT1 for NRF2. To the best of our knowledge, NRF2 overexpression does not induce SIRT1. Our stress resistance data also indicate a definitive upstream role for SIR-2.1 in protection against rotenone (Fig 4h), while SKN-1 plays a contributory role.

Figure 4 convincingly demonstrates an impact of hydralazine on multiple metabolic pathways. Does hydralazine also impact systemic glucose levels? Is there evidence for an impact of hydralazine on blood glucose levels and metabolic indicators in diabetic patients?

PKA is one of the known downstream effectors of beta-adrenergic signaling which upon activation affects systemic glucose. Thus, it seems likely that there is some affect. We agree that this is an interesting question, but found no information on diabetic patients in the literature to provide an answer.

Supplementary Figure 1. Lack of an impact of HIF1 on hydralazine-induced SIRT1 signaling is somewhat unexpected. What is the impact of hydralazine in the *C. elegans* system? Is this cell culture system sufficient to study HIF1α?

The systemic impact of HIF1A on hydralazine-induced SIRT1 in *C. elegans* was studied using mutant animals lacking a functional HIF1A. As demonstrated in

supplementary Figure 1f, we did not observe any link between HIF1A and hydralazine-mediated protection against rotenone, which we showed is completely SIRT1-dependent.

Reviewer #3 (Remarks to the Author):

Dr. Dehghan and colleagues showed that hydralazine (a FDA approved drug) has anti-aging properties via the activation of the NRF2/SKN-1 signaling pathway. In a previous study (Dehghan E. et al. Hydralazine induces stress resistance and extends *C. elegans* lifespan by activating the NRF2/SKN-1 signaling pathway. Nat Commun 8(1):2223, 2017). The present work continues and deepens the molecular and metabolic mechanisms of hydralazine. The authors show that hydralazine improves mitochondrial function and metabolism in a SIRT1/SIRT5-dependent manner. Moreover, they described that this molecule is able to protect *C. elegans* against high glucose toxicity. The present manuscript underlines that the mitochondrial effects of hydralazine are essentials for its anti-aging properties.

Although this manuscript describes several interesting findings, some major and minor points should be addressed (listed below in no particular order).

1. The cell death and cell viability evaluation should be corroborated:
(a) Figures 1a, 2e, 2f, 3a, 6b: the authors should evaluate cell death (PI or DAPI staining) in these experimental conditions.

We have several observations that indicate little or no effect on cell death. We measured mitochondrial membrane potential by either flow cytometry or microplate reader. Based on our viability data, we know that concentrations below 10 μ M, which we used for most of our assays, have no adverse effect on cell viability and even protect from stress-induced cell death. To address the reviewer's concern directly we performed an independent FCM analysis using DAPI and did not observe any increase in the number of dead cells. Elevated mitochondrial membrane potential in *C. elegans* also support no adverse effects on organism health.

2. Figure 3b: the cell viability has been evaluated with the CellTiter-Glo assay (Promega, G7571) (pg. 40, lines 4-11). This method determines the number of viable cells in culture based on quantitation of the ATP present (as an indicator of metabolically active cells). Because the authors demonstrated that Hydralazine modulates SIRT1 and SIRT5 and activates mitochondria, they should evaluate the effect on viability with an assay independent to mitochondria.

We have moved the panel that showed cell viability to the supplementary figures (supplementary Fig. 3b) to avoid any confusion. In our previous paper, we used an MTT cell viability assay and observed a similar effect (PMC5738364).

3. Discussion paragraph: The authors should take into account the paper of Ruiz-Magana MJ et al. "The antihypertensive drug hydralazine activates the intrinsic pathway of apoptosis and causes DNA damage in leukemic T cells. *Oncotarget*. 2016;7(16):21875-86. doi: 10.18632/oncotarget.7871.

Thank you for bringing this study to our attention. We now include it in the discussion (page 22, lines 13-15).

4. Figures 1b, 1d-1g, 1i, 2g: internal positive and negative controls are missing in these experimental settings.

We did use both positive and negative controls in our initial critical experiments to validate our hypothesis and optimize our methods. We understand controls are essential but the choice of controls is equally important. Even though we used isoniazid and resveratrol (Fig 1a and 1c), both these compounds are known to activate multiple pathways so we did not deem them suitable for the remaining experiments. We screened compounds similar to hydralazine but did not find a true negative control.

5. In my opinion, in order to simplify the reading of the manuscript, the figures should be made in a homogeneous manner. For example, in figure 1a, the data are shown as "Mitochondrial membrane potential (% Ctrl)", whereas the same type of experiment is shown as "mitochondrial membrane potential (RFU)" in figures 2e and 2f. Often, the ordinate axis is different for similar comparable experimental settings (i.e. figures 1d-1g or figures 4d-e/figure 5e).

We have made the requested changes by converting all mitochondrial membrane potential values shown in RFU to % Ctrl. The ordinate axis selected to best show the differences between values. Making ordinate axis all the same will result in compression of the data and more difficult visualization of the signal differences. We generated figures with similar ordinate axis but after visual comparison of both, decided to keep the original ordinate axis.

6. Legend of figure 1h: the description of bars graph of figure 1h is missing (but probably is linked with legend of figure 1i?).

The statistical values were missing for figure 1h which are now provided.

7. Results paragraph (lines 12-14) concerning figure 2h: a mitochondrial marker, cytochrome c oxidase subunit IV (COXIV) is described in the text but it is missing in the figure.

The marker was referred to in error and has now been removed from the text.

8. Figures 2b/2h: In Figure 2h, in "Ctrl(shRNA)" condition, 10 uM of hydralazine treatment increases SIRT1 levels in a very significant manner, however in figure 2b the effect of the same concentration of hydralazine in normal cells is not so evident (at least in the representative SIRT1 w. blot shown). The time of Hydralazine (10uM) treatment is the same in figures 2b and 2h? (This information is missing in the figure legend).

We apologize for omitting this piece of information. In Fig 2b, the experiment was for 24 h, while in panel 2h the experiment was for 48 h. Corrected now.

9. In general, it is not clear, for all the w. blot experiments, how the normalization (vs. the loading control) has been done. Which is the meaning of “Relative protein abundance (% Ctrl)” (the information is missing in the Materials & Methods paragraph)? Particularly, in figure 5f the actin bands shown are not convincing, whereas GFP levels are the same in all the experimental conditions. We presented the relative amount of proteins quantified by Western blot analysis as “protein abundance”. The intensity of each protein was normalized to the corresponding internal control (actin or GAPDH), then the normalized values from treatments were divided by the corresponding control(s) to calculate the relative abundance of each protein. This description has been added to the material and method, now. The purpose of using mitochondrial GFP-tagged strain was to study the effect of hydralazine on mitochondrial morphology. The GFP western blot was shown to demonstrate hydralazine-mediated increase in mitochondrial mass which was not supporting the morphology study. The GFP blot was removed from our revised manuscript to prevent confusion.

10. Figure 3d: The AN cells (as AP cells) have been treated with Hydralazine? Moreover, the COXIV w. blot shown is not so representative of the sentence “key regulatory elements of mitochondrial function..... were adversely affected by tau aggregates” (the levels of COXIV in AN and AP in control conditions seem the same). We apologize but our figure was mislabeled in the original manuscript which now has been corrected. The first two lane are blots from AN cells and the next four lanes are blots from AP cells (first two lane from AP cells are from untreated cells and the next two are from cells treated with 5 uM hydralazine). COXIV was initially included as a marker of mitochondrial mass but we had a hard time with acquiring consistent blots for this particular marker. Because we had other markers supporting increase in mitochondrial mass and function (i.e. western blot of CYCS and ATPB), mtDNA/nDNA (data not shown), and ATP (not shown) in AN and AP cells we did not see need to include COXIV WB. If the reviewer would like, we can incorporate the data in our manuscript quickly as the data is already acquired and processed.

11. Figure 3f: In the graph, a protection was observed against rotenone in AAK-2 mutants treated with hydralazine and not as written in the text (p. 11 lines 2-3) “We only observed rotenone resistance in SIR-2.1 mutant worms (Fig. 3f)”. The authors should clarify this point. We agree with the reviewer’s comment. This section of the text was confusing and incomplete so it has been replaced with the following “To deconvolute the role of SIRT1 from AMPK in rotenone resistance, we used SIR-2.1 and AAK-2 (AMPK orthologue in *C. elegans*) double mutant and SIR-2.1 and AAK-2 single mutants. We only observed rotenone resistance in wild-type and AAK-2 mutant

worms indicating that hydralazine-mediated protection is AMPK-independent (Fig. 4h)

12. Figure 4i: the authors should describe in the text the (significant) results obtained with the *skn-1(zu67)* mutant *C. elegans* as well as the meaning of the utilization of several *skn-1* mutants.

We have rewritten this section explaining the rationale for using these mutants and the conclusions we reached using these mutants (Page 11 and page 21-22). The main goal in using *skn-1(zu67)* mutant *C. elegans* was to understand the correlation between hydralazine-mediated mitochondrial activation and extension of lifespan. We show that *C. elegans* with loss of function mutation in all SKN-1 isoforms (*skn-1(zu135)*) which did not experience lifespan extension with hydralazine treatment also did not show mitochondrial activation. However congruent with the essential role of ASI neuronal SKN-1 (*skn-1(zu67)*) (but not intestinal SKN-1 (*skn-1(zu135)gels10*) in extension of lifespan¹⁸, presence of neuronal SKN-1 was enough to promote mitochondrial function with hydralazine treatment.

13. Legend of figure 5f: the size bar meaning is missing.

Due to significant changes in the manuscript as a result of hydralazine target discovery and reasons that we explained before (Reviewer#3, Q9) regarding the purpose of this figure, it was removed.

14. Typo “improveed”, p.16 line 13

Thank you for noting the typo. We have corrected it.

15. Legend of figure 6e: the legend of this part of the figure should be improved.

Thank you for the comment. We have rewritten the legend and hope that clarity is significantly better.

16. A statistic paragraph should be added to the manuscript.

We have now explained the applied statistics for all the experiments in the figure legends. As you suggested, we have also added a paragraph on statistics in the Materials and Methods.

Reviewers' Comments:

Reviewer #1:

Remarks to the Author:

The authors have addressed most of my comments and added exciting new data indicating that PKA is likely a direct target of hydralazine. This increases potential impact of the paper and, in principle, I feel it could now become suitable for Nat Commun. However, the authors should be more cautious with drawing conclusions from their data for PKA (also see specific comments to Figure 5 below). I think the data indicate that PKA is a target of hydralazine, but there could be others, given the high hydralazine concentrations used for some of the assays.

In addition, it appears that the manuscript was not properly proof read before submission (or that a non-final version was accidentally submitted), given the large number of nonsensical text fragments and spurious yellow highlighting of incompletely edited sections.

Abstract:

Instead of highlighting their own recent finding, the authors should highlight the sustained importance of hydralazine in medicine.

Main text:

There are several sections that suggest that the uploaded version still contained tracked changes, and other unintended elements, e.g. yellow highlighting of incompletely revised text. Further, there are multiple paragraphs with sentence fragments or odd repetitions, for example:

- line 345-346
- line 305-312: here, it seems as if two different versions of text were unintentionally retained.
- line 271
- line 281-282
- line 223-229 yellow highlighted region: this sentence is incomprehensible, probably was highlighted for this reason, but then not changed?
- line 66 and following paragraph: many spelling and grammar errors (e.g. "to increases")

Discussion:

- page 22, "explanation for the complementary effect of intestinal SKN-1". It is unclear what is meant here – where is a complementary effect of intestinal SKN-1 demonstrated?
- line 493: "but is not" should be "but are not"
- line 496-502: this entire section is very confusing. Here, the PKA, sirtuin, NRF2 cascade is presented not as causal to, but rather as phenomena separate from the protection against stressors by hydralazine.

Figures:

- the melting point curves in Figure 5 a (the "a" is missing from the Figure) seem quite noisy. How often was this experiment repeated?
- The blots in Figure 5b should be adjusted for contrast – there's hardly a difference between 0 and 100 uM of hydralazine.
- in Figure 5g the legend in the Figure "0", "5", "10" is missing something – likely "uM Hyd".
- the entire legend for Figure 5b needs attention. There are many formatting and other mistakes. Especially the use of italics is not consistent and confusing. This also applies to some of the other figure legends.

Reviewer #2:

Remarks to the Author:

The authors addressed my previous experimental issues to satisfaction. Identification of PKA as upstream hydralazine target is a plausible unifying mechanism. Albeit it is not clear at the mechanistic level how hydralazine impacts PKA.

At the conceptual level the authors did not provide a satisfying explanation of why the a beneficial effect of hydralazine on metabolism has never been reported. The authors' response rather adds to my concern that reported effect is not biologically relevant in patients. This concern could be diffused by analysis of few clinical samples.

Reviewer #3:

Remarks to the Author:

The authors answered to my questions (generally removing the "confusing results") and the manuscript is really improved.

Reviewers' comments:

Reviewer #1 (Remarks to the Author):

The authors have addressed most of my comments and added exciting new data indicating that PKA is likely a direct target of hydralazine. This increases potential impact of the paper and, in principle, I feel it could now become suitable for Nat Commun. However, the authors should be more cautious with drawing conclusions from their data for PKA (also see specific comments to Figure 5 below). I think the data indicate that PKA is a target of hydralazine, but there could be others, given the high hydralazine concentrations used for some of the assays.

In addition, it appears that the manuscript was not properly proof read before submission (or that a non-final version was accidentally submitted), given the large number of nonsensical text fragments and spurious yellow highlighting of incompletely edited sections.

We apologize for the confusion about the text and have proofread the manuscript eliminating the tracking of revisions. We highlighted the text to inform the editor of the changes that we made on the second submission. Our current revised manuscript has all revised sections highlighted upon the editor's request.

Abstract:

Instead of highlighting their own recent finding, the authors should highlight the sustained importance of hydralazine in medicine.

We rewrote the abstract to give a bird's-eye view of why and where our findings are important. We eliminated some of the details of our findings and focused on the big picture instead, as requested.

Main text:

Several sections suggest that the uploaded version still contained tracked changes, and other unintended elements, e.g. yellow highlighting of incompletely revised text. Further, there are multiple paragraphs with sentence fragments or odd repetitions, for example:

- line 345-346

We think many of the problems came from tracking changes which have been eliminated in this draft. Clarifications follow each of the designations of confusing text.

For lines 345-346

We have now broken the first sentence, that was four lines long, into two sentences to make it easier to read and understand.

- line 305-312: here, it seems as if two different versions of the text were unintentionally retained.

The following redundant sentences were deleted from the manuscript. "To explore the possible role of AMPK and SIRT1 pathways on rotenone stress resistance, we used double mutant *C. elegans* impaired in both regulatory elements. While strong protection was observed against a lethal dose of rotenone (50 μ M) in wild-type nematodes treated with hydralazine, double mutant animals failed to show any protection (Fig. 4h)."

- line 271

These two fragmented sentences - "1) The pro-aggregant transgenic *C. elegans* strain pan-neuronally expresses a highly amyloidogenic mutated F3 Δ K280 fragment of human tau driven by the rab-3 promoter. 2) was used this strain as an *in vivo* tauopathy model." - were emerged into a more coherent single sentence; "A pro-aggregant transgenic *C. elegans* strain which pan-neuronally expresses a rab-3 promoter-driven, highly amyloidogenic, mutated F3 Δ K280 fragment of human tau, was used as an *in vivo* tauopathy model."

- line 281-282

The highlighted phrase was removed.

- line 223-229 yellow highlighted region: this sentence is incomprehensible, probably was highlighted for this reason, but then not changed?

We changed the wording in sentences spanning lines 223-229 to improve comprehensibility.

"As expected, *C. elegans* with loss of function mutation in all SKN-1 isoforms (*skn-1* (*zu135*)) which did not experience lifespan extension with hydralazine treatment also did

not show mitochondrial activation (Fig. 3i). However congruent with the essential role of ASI neuronal SKN-1(*skn-1 (zu67)*) (but not intestinal SKN-1(*skn-1 (zu135)/gels10*)) in the extension of lifespan, presence of neuronal SKN-1 was enough to promote mitochondrial function with hydralazine treatment (Fig 3i).”

Was changed to:

“As expected, *C. elegans* strain *skn-1 (zu135)* with loss of function mutation in all SKN-1 isoforms that did not experience hydralazine-mediated lifespan extension, also did not show mitochondrial activation with hydralazine (Fig. 3i). Congruent with the essential role of ASI neuronal SKN-1 in the extension of lifespan, the presence of neuronal SKN-1 in *skn-1 (zu67)* strain recapitulated the beneficial effects of hydralazine on mitochondrial function (Fig 3i).”

- line 66 and following paragraph: many spelling and grammar errors (e.g. “to increases”)

The “s” was removed from “increases”. We could not find any other spelling and grammatical errors in that paragraph.

Discussion:

- page 22, “explanation for the complementary effect of intestinal SKN-1”. It is unclear what is meant here – where is a complementary effect of intestinal SKN-1 demonstrated?

We showed in our previous publication that ASI neuronal SKN-1 is necessary for hydralazine-mediated lifespan extension while the presence of intestinal SKN-1 has a very mild effect. We propose here that the mild hydralazine-mediated lifespan extension observed in strains with intestinal SKN-1 might be due to the role it plays in ROS homeostasis that would be perturbed by mitochondrial activation. We tried to clarify this point in the revised manuscript. This sentence was added for further clarification “The activation of intestinal SKN-1 with hydralazine will decrease the ROS generated by activated mitochondria which is known to have an adverse effect on health and lifespan.”

- line 493: “but is not” should be “but are not”

“is” was changed to “are”.

- line 496-502: this entire section is very confusing. Here, the PKA, sirtuin, NRF2 cascade is presented not as causal to, but rather as phenomena separate from the protection against stressors by hydralazine.

“The activation of PKA, sirtuins and NRF2 point” was combined with “protection against various stressors point” to emphasize that the latter effects are the result of the former activations in the following sentence “activation of PKA, sirtuins and NRF2 known to have profound impact on mitochondrial function, antioxidant homeostasis, and cellular health and stress protection which deteriorates with age”

Figures:

- the melting point curves in Figure 5 a (the “a” is missing from the Figure) seem quite noisy. How often was this experiment repeated?

Thermal Proteome Profiling (TPP) was used as an unbiased screening tool to find hydralazine’s candidate target protein(s). This is a single complex experiment, results of which have been validated using multiple lines of evidence. This technique uses cellular a thermal shift assay with a null hypothesis that either a drug stabilizes or destabilizes proteins it binds in the context of a complex proteome. The melting curves showed in figure five passed multiple statistical criteria for normalization, melting curve fitting and significance of the shift. For example, the number of data points in each condition, minimum slope in the control vs. treatment, the correlation coefficient of the curve, the plateau of the fitted melting curve, etc. were calculated to find if a change in the thermal stability of each PKA subunit was caused by hydralazine or were random. It is noteworthy that both subunits of PKA were significantly affected (in opposite directions) by hydralazine which makes it even more unlikely for the PKA subunit shifts to be artifacts. Moreover, as western blot analysis in Fig. 5b shows, the effects of hydralazine on the stability of PKA is significant. The effect of hydralazine on the activity of PKA and overall biological significance of the interaction were also confirmed using independent biochemical techniques.

- The blots in Figure 5b should be adjusted for contrast – there’s hardly a difference between 0 and 100 uM of hydralazine.

The resolution of these blots was probably affected by multiple conversions from PPT to PDF and again to PDF by the journal website. The original blots are shown below for reviewer examination. We will submit a higher resolution blots to address this particular concern.

- in Figure 5g the legend in the Figure “0”, “5”, “10” is missing something – likely “uM Hyd”.

The legend was revised as follows, “quantification of TMRE signal by microplate reader demonstrating a PKA-dependent elevation in $\Delta\psi_m$ in HEK293 cells treated with 0, 5, and 10 μ M hydralazine for 48 h.”

- the entire legend for Figure 5b needs attention. There are many formatting and other mistakes. Especially the use of italics is not consistent and confusing. This also applies to some of the other figure legends.

We corrected the entire legend for Figure b and made all statistical sentences consistently italic. Please see in text.

Reviewer #2 (Remarks to the Author):

The authors addressed my previous experimental issues to satisfaction. Identification of PKA as upstream hydralazine target is a plausible unifying mechanism. Albeit it is not clear at the mechanistic level how hydralazine impacts PKA.

At the conceptual level, the authors did not provide a satisfying explanation of why the beneficial effect of hydralazine on metabolism has never been reported. The authors’ response rather adds to my concern that reported effect is not biologically relevant in patients. This concern could be diffused by analysis of few clinical samples.

We thank the reviewer for this point to encourage us to include a more detailed account of likely metabolic actions in previously published studies, as elaborated on at the end of this response, as well as reasons why its metabolic actions may not be apparent from patient data, discussed next. In this regard, we think that the analysis of a few clinical samples will not have sufficient statistical power to draw any meaningful conclusions. A clinical trial is being planned but will take more than 2 years in order to write the trial, pass IRB approval, enroll patients and analyze the results. Thus, it is clearly beyond the scope of this study.

To the best of our knowledge, there has not been any clinical trial to specifically look at the possible effects of hydralazine on metabolism. It is possible that the metabolic effects of hydralazine have remained unnoticed due to high dosage administration combined with its usual short-term usage in acute cases. Our data in figure 6

demonstrate the dosage sensitivity of the response. Additionally, the effects of hydralazine on metabolism are slow in onset. Responses require a few days to be detectable. Furthermore, because hydralazine is usually used in combination with other medications in cases of heart failure and complicated diabetes mellitus, any observed effects could not have been attributed to hydralazine alone.

One metabolic effect of hydralazine not seen with other antihypertensive agents is its ability to lower blood lipids. Interestingly hydralazine treatment has been documented to significantly reduce plasma cholesterol in human patients by an unexplained mechanism¹.

The beneficial effects of hydralazine on metabolites affected by diabetes are documented in streptozotocin-induced diabetic rats². Hydralazine-treated rats gained weight more slowly than the control group over a six-week period. A significant reduction in serum triglycerides, phospholipids, free fatty acids and cholesterol have also been reported².

The beneficial effects of hydralazine on a few parameters related to metabolism in diabetic mice have been reported³. Hydralazine sharply reduced the elevated levels of blood urea nitrogen, creatinine, cholesterol and triglycerides³.

In a very recent mouse study, administration of hydralazine for three weeks has been reported to be beneficial in MPTP-induced Parkinson's model. Hydralazine was shown to protect dopaminergic neurons and improved recovery of locomotor performance of the animals⁴. These effects are suggested to be mediated via activation of the NRF2 pathway in this PD mouse model.

We have added a distilled version of the response above in the discussion section of our manuscript. We believe it would be accurate to say that orally administered hydralazine is an older drug that has largely been supplanted by other anti-hypertensives for daily use. Hydralazine is usually administered i.v. for acute control of elevated blood pressure. Obviously, this is given over a shorter course and it is probably not long enough to assess for metabolic effects, especially in acutely ill patients who usually have multiple co-morbidities.

Reviewer #3 (Remarks to the Author):

The authors answered to my questions (generally removing the “confusing results”) and the manuscript is really improved.

We thank reviewer #3 for reading our manuscript again and supporting its publication.

References:

- 1 Deming, Q. B., Hodes, M. E., Baltazar, A., Edreira, J. G. & Torosdag, S. The changes in concentration of cholesterol in the serum of hypertensive patients during antihypertensive therapy. *The American journal of medicine* 24, 882-892 (1958).
- 2 Rodrigues, B., Goyal, R. K. & McNeill, J. H. Effects of hydralazine on streptozotocin-induced diabetic rats: prevention of hyperlipidemia and improvement in cardiac function. *The Journal of pharmacology and experimental therapeutics* 237, 292-299 (1986).
- 3 Kesavan, S. K. et al. Proteome wide reduction in AGE modification in streptozotocin induced diabetic mice by hydralazine mediated transglycation. *Scientific reports* 3, 2941, doi:10.1038/srep02941 (2013).
- 4 Guo, X. et al. Hydralazine Protects Nigrostriatal Dopaminergic Neurons From MPP(+) and MPTP Induced Neurotoxicity: Roles of Nrf2-ARE Signaling Pathway. *Frontiers in neurology* 10, 271, doi:10.3389/fneur.2019.00271 (2019).